# Super-enhancer-guided mapping of regulatory networks controlling mouse trophoblast stem cells

Bum-Kyu Lee[1,2], Yu jin Jang [1], Mijeong Kim [1], Lucy LeBlanc[1,2], Catherine Rhee[1], Jiwoon Lee[1], Samuel Beck[1], Wenwen Shen[1] & Jonghwan Kim [1,2,3]*

Trophectoderm (TE) lineage development is pivotal for proper implantation, placentation, and healthy pregnancy. However, only a few TE-specific transcription factors (TFs) have been systematically characterized, hindering our understanding of the process. To elucidate regulatory mechanisms underlying TE development, here we map super-enhancers (SEs) in trophoblast stem cells (TSCs) as a model. We find both prominent TE-specific master TFs (Cdx2, Gata3, and Tead4), and >150 TFs that had not been previously implicated in TE lineage, that are SE-associated. Mapping targets of 27 SE-predicted TFs reveals a highly intertwined transcriptional regulatory circuitry. Intriguingly, SE-predicted TFs show 4 distinct expression patterns with dynamic alterations of their targets during TSC differentiation. Furthermore, depletion of a subset of TFs results in dysregulation of the markers for specialized cell types in placenta, suggesting a role during TE differentiation. Collectively, we characterize an expanded TE-specific regulatory network, providing a framework for understanding TE lineage development and placentation.

[1] Department of Molecular Biosciences, The University of Texas at Austin, Austin, TX 78712, USA. [2] Institute for Cellular and Molecular Biology, The University of Texas at Austin, Austin, TX 78712, USA. [3] Center for Systems and Synthetic Biology, The University of Texas at Austin, Austin, TX 78712, USA. *email: jonghwankim@mail.utexas.edu

The placenta is a temporary yet crucial organ that facilitates nutrient uptake, waste elimination, and gas exchange through the mother's blood supply to maintain a healthy pregnancy[1]. The origin of the placenta is the trophectoderm (TE), an outer layer of the blastocyst stage of embryo. Placental tissues comprises multiple specialized cell types such as trophoblast giant cells (TGCs), spongiotrophoblasts (SpTs), and syncytiotrophoblasts (SynTs)[1]. Abnormal TE lineage differentiation is a major cause of pregnancy complications[2,3].

In contrast to embryonic stem cells (ESCs), derived from the inner cell mass[4] and intensively researched due to their pluripotency, trophoblast stem cells (TSCs) established from an outgrowth of either polar TE or extra-embryonic ectoderm[5] have not been studied nearly as much, despite the importance of the placenta. This paucity of research has led to only a rudimentary understanding of the mechanisms underlying TE lineage specification, maintenance, and differentiation. During the past decade, rodent TSC models have been utilized to study placenta development[6]. Whether human TSCs recently established truly constitute a tractable model for placental development awaits further confirmation[7].

Cell-type-specific transcription factors (TFs) and their target cis-regulatory elements, mainly enhancers, orchestrate lineage-specific gene expression programs to determine cellular identity and functions[8]. In TE lineage, only a few TFs have been identified as key regulators, such as Tead4, Cdx2, Gata3, Tfap2c, Eomes, Ets2, and Elf5[9–15]. The precise transcriptional regulatory mechanisms of TE/TSC-specific TFs remain elusive. There has been no thorough investigation in how an expanded set of TFs, including previously uncharacterized factors, controls self-renewal of TSCs, modulates differentiation toward more specialized cell types, or interacts with their respective chromosomal targets via forming regulatory networks with their partner proteins, including epigenetic regulators.

The concept of "super-enhancers" (SEs), consisting of a cluster of enhancers, has been recently proposed[16,17]. They can be predicted by strong occupancy signals of mediators, p300 (Ep300), or enhancer-specific histone modification marks, such as H3K27ac and H3K4me1. In ESCs, SEs are associated with master pluripotency TFs, such as Oct4 (Pou5f1), Sox2, and Nanog. In parallel, multiple master TFs co-occupy common SEs, further amplifying the levels of SE-associated target genes including master TFs themselves by forming highly intertwined regulatory circuitry. While previous studies attempted to identify global enhancer usage in TSCs or placenta[18–21], there have been no systematic approaches to predict master regulators by defining enhancers or SEs.

In this study, we identify TSC-specific SEs and subsequently predict putative key TFs. SE-associated genes we defined in TSCs are implicated in placental development, and many of these encode for TFs, including almost all known TE/TSC-specific TFs as well as numerous TFs that have never been previously implicated in placental biology. By mapping global target loci of over two dozen known and newly identified SE-predicted TFs, we reveal that they not only co-occupy various target genes, most of which are highly active in TSCs and the placenta, but also form an intricate regulatory network. Accordingly, some TFs show dynamic switches in their enhancer-binding patterns upon differentiation. Interestingly, the SE-associated TFs show four distinct expression patterns during TSC differentiation and these different classes of TFs demonstrate discrete roles during TE lineage differentiation. Our findings will serve as a valuable resource for research in placental development and disorders.

## Results

**Mapping of TSC-specific enhancers and SEs.** To identify enhancers utilized in TSCs, we first mapped the genomic occupancy of p300 using chromatin immunoprecipitation coupled with next-generation sequencing (ChIP-seq). As a control, ChIP-seq in ESCs was performed in parallel. We identified total 39,957 and 36,190 enhancers in TSCs and ESCs, respectively (Supplementary Fig. 1a, b, Supplementary Data 1, 2). The majority of the p300-binding sites were distal from the transcriptional start sites (TSSs) of well-annotated genes as expected (Supplementary Fig. 1c) and ~74% and ~75% of enhancers in TSCs and ESCs, respectively, were cell-type-specific (Fig. 1a, Supplementary Fig. 1d). As shown in Fig. 1b, previously known TE-specific TFs, such as Elf5 and Gata3, were associated with multiple p300 target sites exclusively in TSCs, whereas the regulatory elements of ESC-specific TFs, such as Oct4 and Nanog, were occupied by p300 only in ESCs. Gene ontology (GO) term analyses revealed that TSC-specific enhancer-associated genes are enriched in placenta and TE development-related terms (Supplementary Fig. 1e).

Then we ranked well-annotated genes in order of relative gene expression (TSCs over ESCs) and plotted the average number of enhancers associated with each gene (Fig. 1c). We observed a strong positive correlation between gene activity and the number of associated enhancers, indicating that multiple cell-type-specific enhancers can synergistically activate the associated target gene. Other enhancer markers—Med12, H3K4me1, and H3K27ac—displayed similar patterns as p300 (Supplementary Fig. 1f), confirming that we have identified bona fide enhancers in TSCs on a global level. Notably, genes with comparable expression levels between TSCs and ESCs (e.g., Esrrb, Tead1, and Trim71) showed differential enhancer usage (Supplementary Fig. 1g), implying that distinct cell-type-specific regulatory machineries control these genes.

While enhancers are generally involved in cell-type-specific gene expression programs[22–24], more recent studies showed that SEs are often associated with cell-type-specific master regulators[16,25]. We identified a total of 1,186 TSC-specific SEs with the criteria previously described[16], then subsequently defined 1,046 SE-associated genes (Fig. 1d, Supplementary Fig. 2a, Supplementary Data 3, 4). Boundaries of SEs were demarcated with the strongest p300 signals (Supplementary Fig. 2b, c), and p300 signal positively correlated with Med12 occupancy, H3K4me1, and H3K27ac marks (Supplementary Fig. 2c). Indeed, compared to regular enhancers (or typical enhancers), SEs harbor broader and stronger p300/Med12/H3K27ac signatures with prominent ATAC-seq (assay for transposase-accessible chromatin using sequencing) signal, linked with greater gene activation (Supplementary Fig. 2d). These affirm that TSC-specific SEs share common features of SEs defined in other contexts[16,17,25].

Notably, the motifs of known TE-specific TFs, such as Gata3, Teads, and Tfap2c, are embedded within the TSC-specific SEs (Supplementary Fig. 2e), implying that SEs may serve as target hubs of multiple TE-specific TFs. GO analysis of SE-associated genes revealed significant enrichment of placenta-associated terms, including embryonic placenta development and trophectodermal cell differentiation, as well as actin cytoskeleton and adherens junction (Fig. 1e. Supplementary Fig. 2f). Importantly, many SE-associated genes were TFs or DNA-binding proteins (Fig. 1e), as well as factors implicated in multiple signaling pathways (e.g., PI3K-Akt, Hippo, and MAPK), implicated in TE lineage development (Supplementary Fig. 2g)[26–28].

**Four different classes of TSC-specific TFs predicted by SEs.** We found that almost all 1,046 SE-associated genes are significantly more active in TSCs compared to ESCs (Supplementary Fig. 2h),

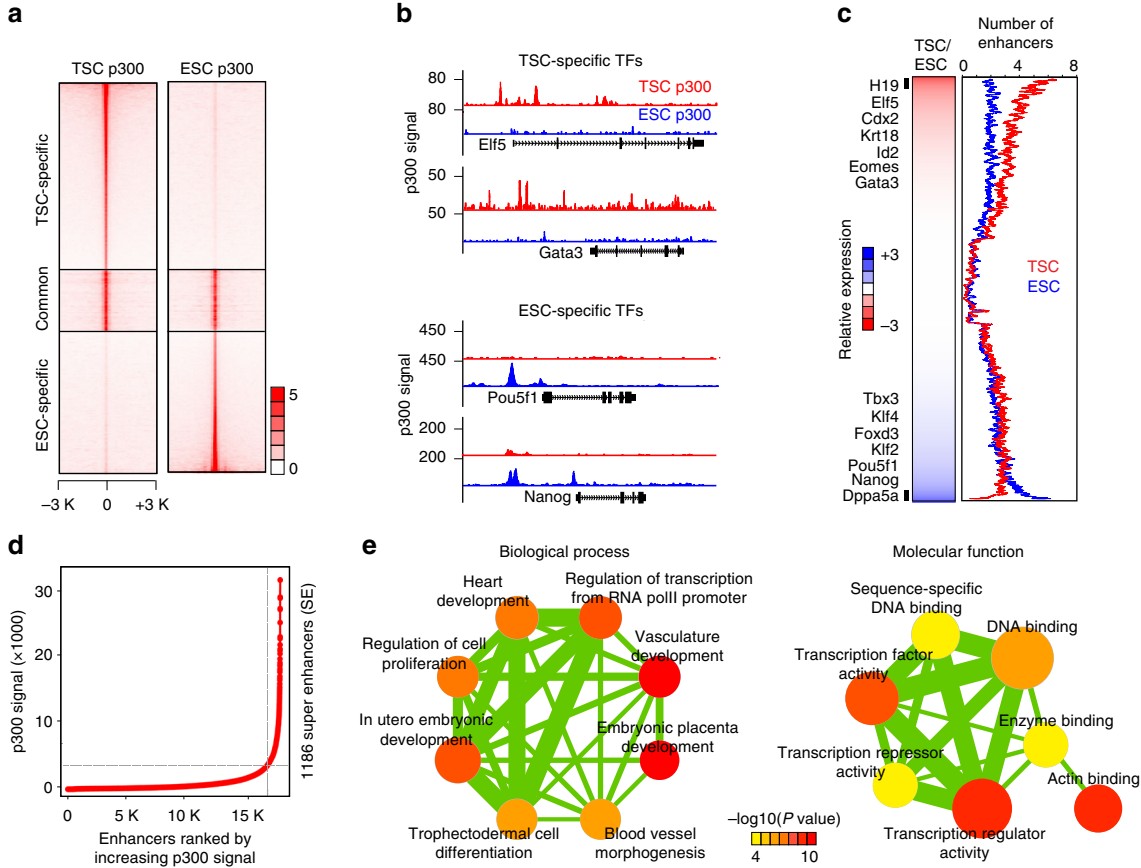

**Fig. 1** TSC-specific enhancers and super enhancers (SEs) are defined. **a** Heatmaps showing occupancy signals of p300 in TSC-specific, common, and ESC-specific p300 sites in TSCs (left) and ESCs (right). **b** Snapshots of p300 ChIP-seq signal tracks around TSC- and ESC-specific genes. **c** A heatmap showing relative gene expression (TSC/ESC) along with a line graph presenting multiple enhancers' association with cell-type-specific genes. Moving average (window size 100) was applied to calculate average number of enhancers associated with each gene. **d** Line graph presenting the number of SEs defined by ranked p300 occupancy signal. **e** Network maps illustrating enriched gene ontology (GO) terms of SE-associated genes in biological process (left panel) and molecular function (right panel). Node color and line thickness indicate *P* value and the extent of overlapped genes between two terms, respectively

and they are expressed substantially more in both the mouse and the human placenta compared to other tissues (Supplementary Fig. 2i). Among SE-associated genes, 197 genes encode sequence-specific TFs, epigenetic regulators, or DNA-binding proteins (hereafter SE-predicted TFs) (Supplementary Data 5). Strikingly, almost all of previously known TFs in TSCs or TE lineage, including Arid3a, Cdx2, Elf5, Esrrb, Gata3, Hand1, Sox2, Tfap2c, and Tead4[10–12,15,27,29–32], were SE-associated factors. This highlights the feasibility of SE-guided mapping of cell-type-specific key TFs. Importantly, our literature search revealed that only about 28% of TFs among 197 SE-predicted TFs have been previously implicated in TE lineage or placenta development (Supplementary Data 6).

To investigate further into their roles in placental development, we analyzed the expression of each SE-predicted TF during time-course differentiation of TSCs. Hierarchical clustering revealed that the TFs fall into four distinct classes based on their expression patterns (Fig. 2a). Class 1 TFs show biphasic expression, and they are most highly expressed in self-renewing and late differentiating TSCs, while Class 2 TFs are gradually downregulated upon differentiation. Class 3 TFs showed gradual upregulation during differentiation, whereas the Class 4 TFs displayed relatively stable expression.

We found that the Class 2 includes mostly known TSC-specific TFs (Elf5, Cdx2, Esrrb, Sox2, and Eomes) involved in proliferation of TSCs or TSC-like progenitors[10,13,30,33–35]. Though highly expressed in TSCs (Fig. 2b), they showed

relatively weak expression levels in the placenta (Fig. 2c), consistent with the time-course differentiation data. This class of TFs may be instrumental to maintain TSC self-renewal or early TE lineage specification. Conversely, most of the Class 3 TFs showed stronger expression in placenta than the Class 2 TFs (Fig. 2c) and they include Arid3a, Cited2, Dlx3, Ets2, and cFos, previously implicated in TE differentiation[14,29,36–39]. Taken together, we identified a plethora of known and unknown SE-predicted TSC-specific TFs displaying dynamic expression patterns during TSC differentiation. This suggests that they play crucial roles at specific stages of placenta development.

**Crucial roles of SE-predicted TFs in healthy placentation.** To elucidate how TE- or TSC-specific properties are transcriptionally controlled, we comprehensively map the chromosomal targets of 28 selected SE-predicted TFs (27 TFs and Ctcf) by ChIP-seq (Fig. 3a, Supplementary Data 5) and identified a total of 155,461 TSC-specific *cis*-regulatory elements (Supplementary Fig. 3a). Individual TFs occupy as many as tens of thousands of targets including genes regulated by promoter–enhancer looping (Supplementary Fig. 3b, Supplementary Data 7) and many SE-predicted TFs tend to co-occupy TSC-specific distal regulatory regions bound by p300 and Med12 (Supplementary Fig. 3c–f), suggesting that the tested TFs promote TSC-specific gene expression programs via

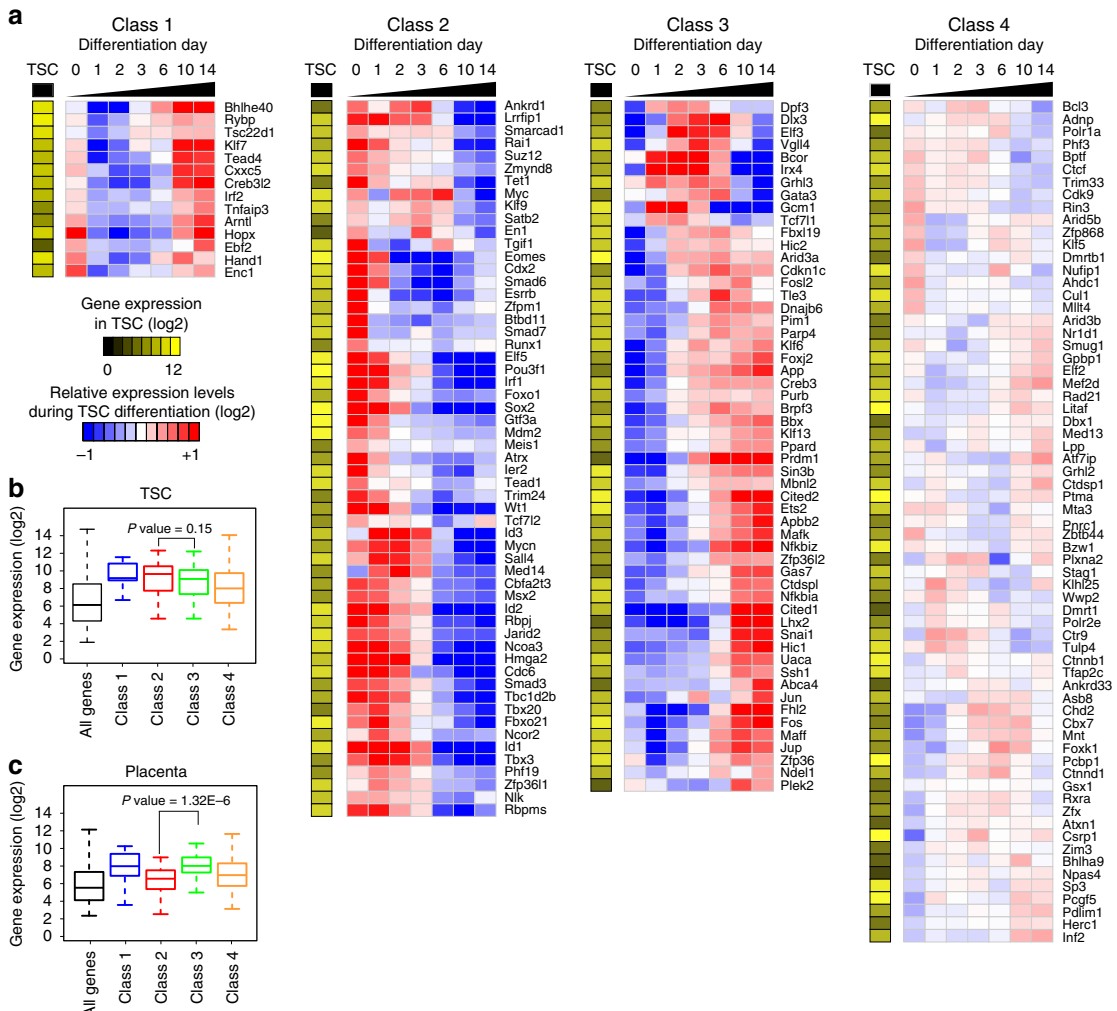

**Fig. 2** Four different classes of SE-associated TFs. **a** Heatmaps illustrating four distinct SE-associated TF groups that show unique expression pattern during time-course differentiation of TSCs. Relative gene expression was calculated by dividing the level of a gene with average gene expression across time points. Absolute gene expression level in self-renewing TSCs was also shown in a single column heatmap beside the time-course heatmaps. **b**, **c** A boxplot showing distribution of expression levels of genes in four classes and all genes as a control in TSCs (**b**) and placenta (**c**). Lower and upper bounds of the box indicate the first and third quartiles, respectively. A horizontal line near the middle of the box shows the median, while whiskers extend from the box to the smallest (largest) datum not further than 1.5 times of the interquartile range. Statistical significance was determined by two-tailed, Wilcoxon test

enhancer binding. As previously reported in diverse cell types[40], Ctcf did not show tissue-specific binding patterns in TSCs (Supplementary Fig. 3c) but demarcates SEs (Supplementary Fig. 3g). Notably, SE-predicted TFs tend to occupy TSC-specific SEs as approximately 95% of SEs was co-occupied by more than 8 TFs (Supplementary Fig. 4a, b) and they are preferentially co-occupied by a group of mainly 14 TFs we tested (Supplementary Fig. 4c, d). Interestingly, these 14 TFs did not belong to any specific class we defined (Fig. 2a), suggesting that, regardless of the class, they modulate TSC self-renewal, perhaps via different mechanisms.

Since improper placental development often results in human pregnancy disorders such as preeclampsia (PE)[2], we reasoned that dysregulation of some SE-associated genes may be associated with human PE. In fact, we confirmed that previously known PE biomarkers, such as FLT, ENG, and BHLHE40[41–43], are TSC-specific SEs-associated factors (Supplementary Data 4). Moreover, by reanalyzing multiple human PE gene expression data sets (see "Methods"), we found >120 SE-associated genes severely dysregulated in human PE placentas compared to healthy placentas (Supplementary Data 8). Many of them have never been characterized in placental disorders.

**TSC-specific transcriptional regulatory network (TRN).** Using the data obtained from ChIP-seq of 27 SE-predicted TFs, we unveiled several interesting features of TSC-specific TRN by visualizing transcriptional interconnectivity of the tested TFs (Fig. 3b). First, the resultant network is composed of the TFs that had known roles in TE lineage development (marked by a round shape) as well as many TFs that had not been characterized in the TE context including Fbxo21, Hic2, Meis1, Maff, Mafk, Pou3f1, and many others (marked by a rectangular shape) (Fig. 3b). Second, our target analysis of 27 SE-predicted TFs unveiled that TSC-specific TFs are involved not only in TSCs but also in various stages of TE development, from specification or maintenance of TE lineage in pre-implantation embryos to further TE lineage differentiation in post-implantation embryos[9–14,29,33] (Supplementary Fig. 4e). Third, the network acts as a hub where both transcriptional regulation and signaling cascades intersect. Notably, the network includes Tead4 and Smad6, which are downstream effectors of Hippo and Tgfβ signaling pathways, respectively, both of which are implicated in TE development or TSC maintenance[27,44]. The TFs in the TRN also co-occupy effectors of other signaling pathways, including Smad3, Smad7, Tgif1, Tead1, and Ctnnb1 (Supplementary Fig. 4f, Supplementary

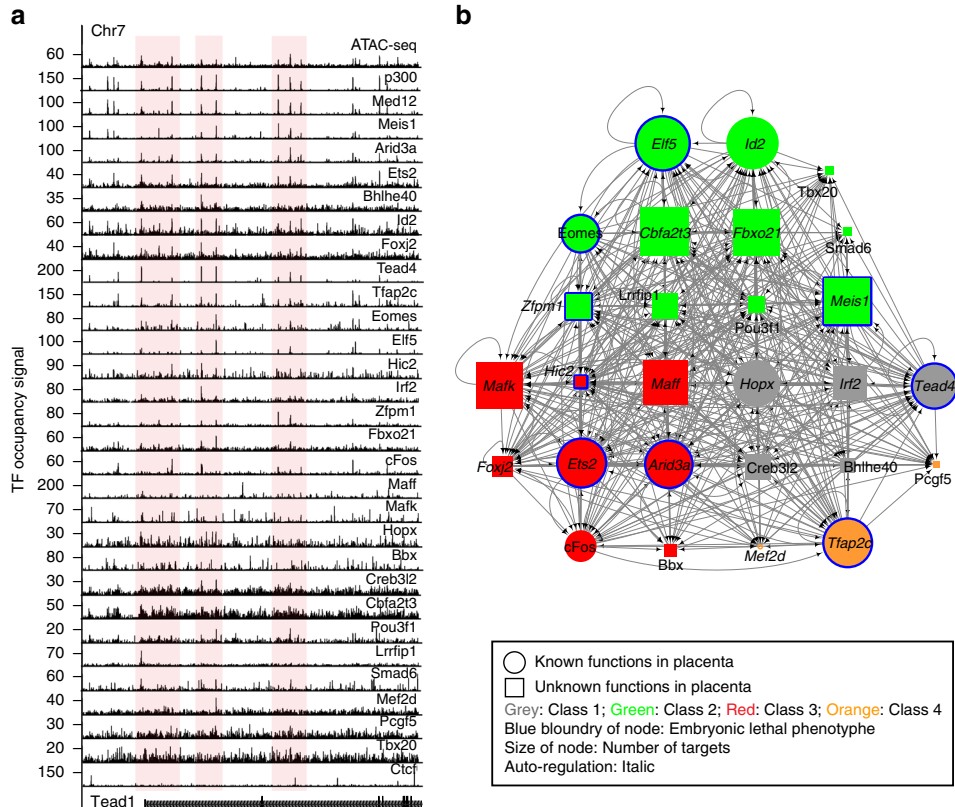

**Fig. 3** TSC-specific transcriptional gene regulatory network. **a** Snapshots of multiple ChIP-seq signal tracks of TFs near Tead1. **b** A transcriptional regulatory network of TSC-specific TFs. TFs having known and unknown functions in TSCs are shown as different shapes (round for known, rectangular for unknown). Classes defined by the expression pattern of a TF during TSC differentiation (Fig. 2a) are shown with different color code. Genes showing a lethal phenotype of embryo upon deletion are shown with blue boundary. TFs showing auto-regulation are shown with italic

Data 4). Fourth, newly defined TSC-specific TFs regulate known TSC-specific TFs and co-occupy the genes robustly implicated in placental and TE development (Supplementary Figs. 3c and 4e). GO analysis of TSC-specific TFs' targets disclosed that their targets are not only associated with abnormal placental phenotypes, including lethality due to faults in placental development in mice (Supplementary Fig. 5a), but also show temporal/differential expression patterns during the various stages of placental ontogeny (Supplementary Fig. 5b). Fifth, the TFs in the network function together to promote gene expression (Supplementary Fig. 5c). Notably, many TFs in the network seem to regulate each other collaboratively rather than hierarchically. Much like the ESC-specific regulatory circuitry[45], when assuming regulator binding to a gene implies regulatory control, TSC-specific TRN implies that most of the TSC-specific-TFs in the network are regulated by feed-forward, feedback, and auto-regulatory mechanisms (Fig. 3b, Supplementary Fig. 5d, e). In total, we established a list of 1,296 target genes that are co-occupied by >22 TFs (Supplementary Data 9). They were preferentially expressed in TSCs over ESCs (Supplementary Fig. 5f), the placenta over other tissues (Supplementary Fig. 5g), and highly enriched with the placenta-related GO terms (Supplementary Fig. 5h). The defined TRN represents a comprehensive TSC/TE-specific nexus of global gene regulation.

**Changes in enhancers and TF binding upon TSC differentiation.** Opposite expression patterns of the TFs in the Class 2 and 3 during TSC differentiation (Fig. 2a) suggested dynamic changes in global gene regulatory modes. To understand this process, we first cataloged the enhancer usage in differentiated TSCs (dTSCs)

(Supplementary Fig. 6a). A plethora of dTSC-specific p300-binding sites that were relatively closed in TSCs emerged in dTSCs in conjunction with loss of TSC-specific p300 sites, indicating that dynamic chromatin remodeling also occurs during TSC differentiation (Supplementary Fig. 6b). MAnorm[46] defined 5,992 statistically significant dTSC-specific enhancers (Supplementary Fig. 6c), and these enhancer-associated genes were enriched in GO terms, such as placenta development, TGC differentiation, and SpT layer development (Supplementary Fig. 6d). Mouse phenotype enrichment analysis revealed that disruption of these genes leads to abnormal development of TGCs and SpTs (Supplementary Fig. 6d). Consistently, dTSC-specific genes such as TGC markers (prolactin and cathepsin gene families) and SpT markers (Tpbpa and Ascl2) were exclusively associated with dTSC-specific enhancers (Fig. 4a). Integrative analysis of the p300 occupancy and gene expression data disclosed that previously known genes implicated in TGC and SpT development are induced upon differentiation along with a drastic increase in or even new appearance of dTSC-associated enhancers (Fig. 4b).

To interrogate to what extent the TFs in 4 different classes (Fig. 2) alter their genomic targets during differentiation, we mapped the targets of total 16 TFs in dTSCs, at least 2 TFs in each class (Supplementary Data 5). Surprisingly, target correlation analysis revealed that most TFs except Class 2 generally show similar occupancy patterns in a context-dependent manner rather than a factor-dependent manner (Fig. 4c). In addition, we revealed that the TFs belonging to different classes behave disparately (Supplementary Fig. 6e). Among four different TF classes, TFs in the Class 2 (such as Elf5, Meis1, and Pou3f1) showed a significantly lower number of dTSC-specific binding sites with weaker occupancy signals, likely due to their reduced

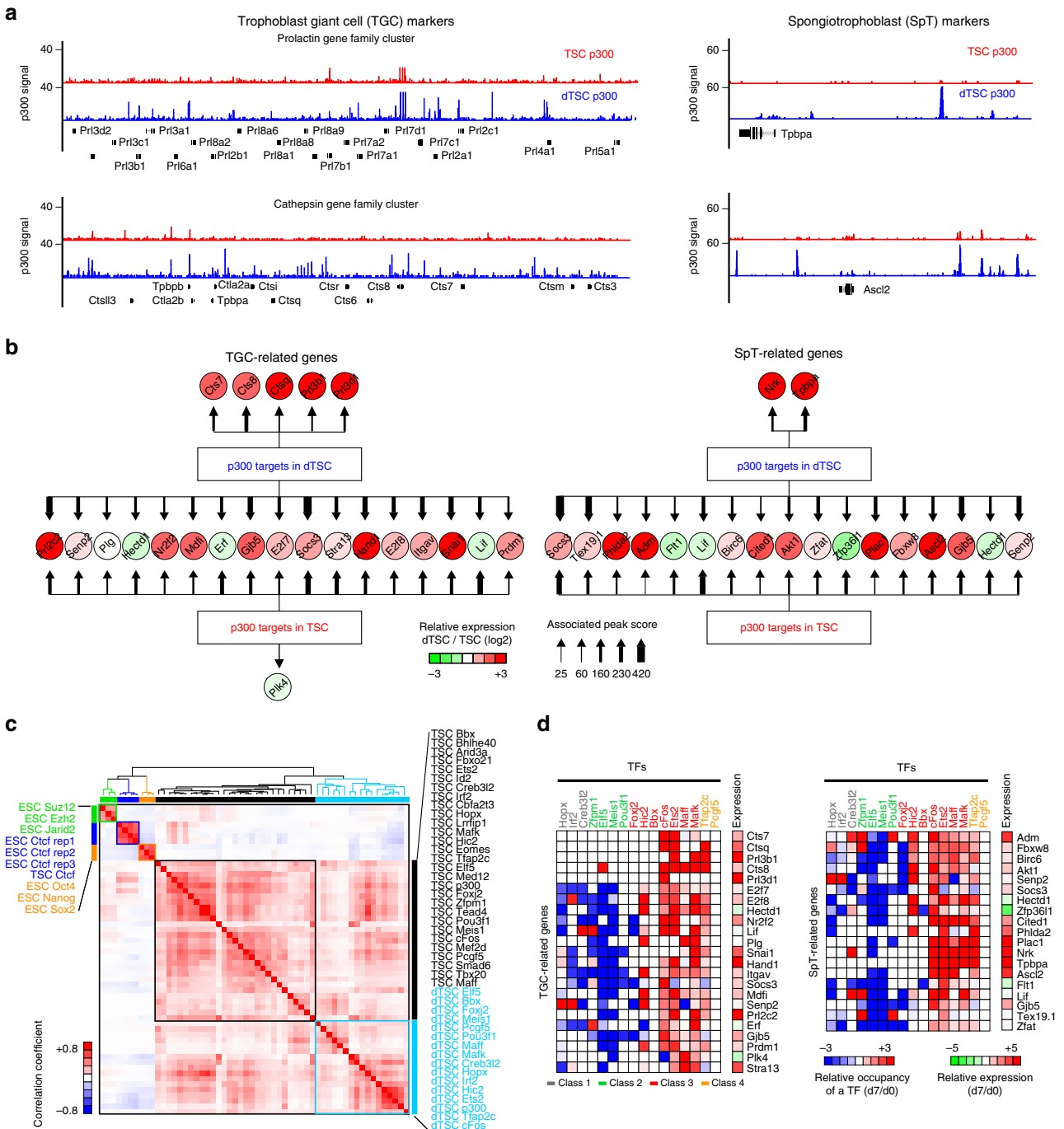

**Fig. 4** Dynamic changes of enhancers' landscape and occupancy of TSC-specific TFs during differentiation of TSCs. **a** Signal track images depicting occupancy of p300 around TGC (left panel) and SpT (right panel) markers in TSCs (red) and dTSCs (blue). **b** p300 occupancy networks of TGC (left panel) and SpT (right panel) marker genes in TSCs and dTSCs. Thickness of an arrow indicates occupancy score of p300. Relative gene expression (dTSCs/TSCs) is shown with color-coded heatmap. **c** A correlation heatmap of various TFs' occupancy. Different functional classes of genes are color coded (Ctcf: blue; pluripotency factors: orange; PRC complex: green; TFs in TSCs: black; TFs in dTSCs: sky blue). Gradient color scale bar on the left indicates correlation coefficients. **d** Heatmaps depicting relative occupancy of TFs (dTSCs/TSCs) in TGC-related (left panel) and SpT-related (right panel) genes. Relative gene expression level (dTSCs/TSCs) is shown right beside the heatmaps

expression upon differentiation (Fig. 2a). Conversely, TFs in other classes largely showed increased occupancy at the dTSC-specific enhancers upon differentiation (Supplementary Fig. 6e, f). Particularly, the Class 3 TFs (cFos, Ets2, Maff, and Mafk) showed increased occupancy at the enhancers associated with TGC- and SpT-related genes in dTSCs (Fig. 4d). In summary, SE-predicted TFs show class-dependent dynamic changes in their target occupancy patterns in a manner aligned with changes in global enhancer usages upon differentiation of TSCs.

**Opposing roles between different classes of TFs.** To characterize SE-predicted TFs in the context of TE lineage development, we performed short hairpin RNA (shRNA)-mediated knockdown

(KD) of 17 TFs (6 previously known: Elf5, Ets2, Hand1, Hopx, Id2, and Tead4; 11 uncharacterized in TE context: Cbfa2t3, Creb3l2, Foxj2, Maff, Mafk, Meis1, Lrrfip1, Pcgf5, Pou3f1, Tbx20, and Zfpm1) followed by differentiation of TSCs for 3 days. Upon confirming at least 80% KD of most of the factors (Supplementary Fig. 6g), we monitored the levels of the marker genes representing specialized TE lineage cell types, such as TGCs (Prl2c2 and others), SpTs (Tpbpa and others), and SynTs (Gcm1, SynA, and SynB). While all tested markers were upregulated after 3 days of differentiation of control TSCs (d3/d0) (Fig. 5a), KD of TFs previously known TSC/TE lineage development either impaired or accelerated marker gene induction as previously reported. For example, deletion of Hand1 inhibits induction of TGC markers[47], whereas depletion of Elf5 or Hopx led to unusually high induction of TGC marker genes[48,49].

Intriguingly, we found that KD of Meis1, Pou3f1, or the majority of other Class 2 TFs (Elf5, Id2, Zfpm1, and Tbx20) generally enhances the levels of TGC/SpT marker genes during TSC differentiation, whereas KD of Maff, Mafk, or other Class 3 TFs dramatically suppresses the activation of multiple TGC/SpT markers during differentiation (Fig. 5b). This suggests that the Class 2 TFs may enhance TSC self-renewal by suppressing differentiation, while the Class 3 TFs facilitate differentiation toward the TGC and SpT fates. This agrees with the expression patterns as Class 2 TFs become downregulated while Class 3 TFs are activated during TSC differentiation (Fig. 2a). We additionally found that syncytiotrophoblast layer II (SynT-II) marker genes (Gcm1 and SynB) seem to show similar expression patterns that are somewhat opposite to syncytiotrophoblast layer I (SynT-I) marker (SynA) upon KD of TSC-specific TFs such as Elf5, Hopx, Id2, Pou3f1, and Zfpm1 (Fig. 5b), which provide a possibility that these TFs might play dual function by suppressing TGC/SpT/SynT-I differentiation while promoting SynT-II differentiation.

To validate this class-dependent functional divergence, we performed global gene expression profiling upon KD of several different TFs including Meis1, Pou3f1, Id2, Lrrfip1, and Zfpm1 (Class 2), as well as Maff, Mafk, Ets2, and Foxj2 (Class 3) upon differentiation. Hopx and Hand1 served as controls, as Hopx depletion leads to substantial propagation of TGC layers with reduction of SpT formation in placenta[49] while deletion of Hand1 prohibits TGC differentiation[50]. First, we intersected targets of differentially expressed genes (DEGs) with our ChIP-seq data and performed GO analysis to understand how and to what extent direct targets of a TSC-specific TF influence differentiation of TSCs. Consistent with our quantitative PCR (qPCR) results (Fig. 5b), depletion of the Class 3 TFs mostly showed enriched placenta development-related GO terms including placenta development, blood vessel development, or cytoskeleton organization in their downregulated target genes. Interestingly, upregulated target genes upon KD of Class 2 TFs were associated with placenta development-related GO terms, such as female pregnancy, blood vessel development, or hematopoiesis (Fig. 5c, Supplementary Fig. 7a, b). These results suggest that TSC-specific TFs directly control the genes involved in placenta development, but their modes of action are largely Class specific.

We further investigated the role of a TSC-specific TFs for lineage specification toward placental cell types using gene set enrichment analysis (GSEA). Intriguingly and similar to the results shown in Fig. 5b, GSEA showed that KD of Maff, Mafk, Ets2, or Foxj2 inhibits induction of SpT- and TGC-related genes, substantiating that each of these seems to be required for SpT and TGC differentiation (Fig. 5d, e and Supplementary Fig. 8a). Conversely, KD of Meis1, Pou3f1, Id2, Pcgf5, Hopx, or Zfpm1 led to stronger activation of TGC-related genes, suggesting that these TFs function as suppressors of TGC differentiation. We conducted additional GSEA using the recently reported gene sets

representing 28 different placental cell types from single-cell RNA-seq analysis[51]. The result consistently supports that Maff, Mafk, Foxj2, or Ets2 KD shows impaired induction of the genes representing invasive SpTs and spiral artery TGCs. On the other hand, Meis1, Pou3f1, Id2, Pcgf5, Hopx, or Zfpm1 KD cells presented opposing patterns (Supplementary Fig. 8b). Taken altogether, we have proposed opposing regulatory roles of TSC-specific TFs during TE lineage development (Fig. 5f).

Finally, we performed immunohistochemistry (IHC) to see whether the functions of TSC-specific TFs proposed by in vitro experiments are correlated with in vivo expression patterns in placenta. We confirmed the area of TGC and SpT in placenta by IHC with proliferin and Tpbpa, respectively (Supplementary Fig. 9a). As shown in Fig. 5g, TFs implicated in TGCs and SpT differentiation, such as Maff, Mafk, Ets2, and Foxj2, showed strong expression in the area of placenta representing TGC and SpT, which further provides another layer of evidence that these TFs play critical roles in the development of TGC and SpT lineages of placenta. Unexpectedly, we also observed weak expression signals of those four genes in the labyrinthine area despite of the confirmed specificity of antibodies used (Supplementary Fig. 9b), implying that they might be also implicated in the development of labyrinthine. On the other hand, Meis1 and Pou3f1 showed weak expression in TGC and SpT lineages (Supplementary Fig. 8c).

## Discussion

Given that the placenta is neglected in research compared to other organs, our data provide a rich and much-needed resource for further exploration of the complex regulatory network modulating normal and abnormal TE lineage development. Remarkably, we identified >150 SE-predicted TFs that have not been systematically characterized in TSCs or TE lineage (Supplementary Data 6). Prior studies suggested that SE-associated genes are often disease linked, as shown in the contexts of T cells[25], tumorigenesis[17], and various genetic disorders[16,52]. In line with this, dysregulated genes in human PE patients including FLT and ENG are SE-associated factors defined in the current study. Our list of SE-associated genes showing abnormal expression in PE placentas (Supplementary Data 8) may serve as critical biomarkers or targets of therapeutic intervention.

Interestingly, while knockout (KO) of a few SE-associated TFs such as Ets2, Eomes, Esrrb, Dlx3, and Hand1[13,14,30,50,53] has known to lead to lethal phenotypes in mice, functional implications in implantation or placentation have not always been reported for KO of many SE-predicted TFs. This might be because many prior studies have focused on fetal phenotypes, leading to mis-annotation of placental phenotypes during early embryo development. Indeed, a recent study has reported that 68% of KO mouse lines that experience lethality during or post mid-gestation display some degree of placental defects, demonstrating how essential it is to take the placenta into account[54]. Moreover, recent re-evaluation of the TFs previously implicated in various developmental processes other than TE lineage, such as KO of Satb2, Prdm1, Arid3a, and Cited2[29,39,55,56], in turn showed defects in placental development. Prudent inspection focusing on TE lineage development should be given for the factors in KO mice models to unveil their uncharacterized roles in implantation and/or placentation. We provide here an extensive catalog of TSC-specific TFs, predicted by SEs. Further functional characterization of these TFs will substantially benefit the fields of placental, developmental, and systems biology.

While all are highly expressed in TSCs, the discovery of multiple classes of SE-associated TFs (Fig. 2) is particularly fascinating. Class 2 and 3 TFs are the most notable, as Class 2 TFs,

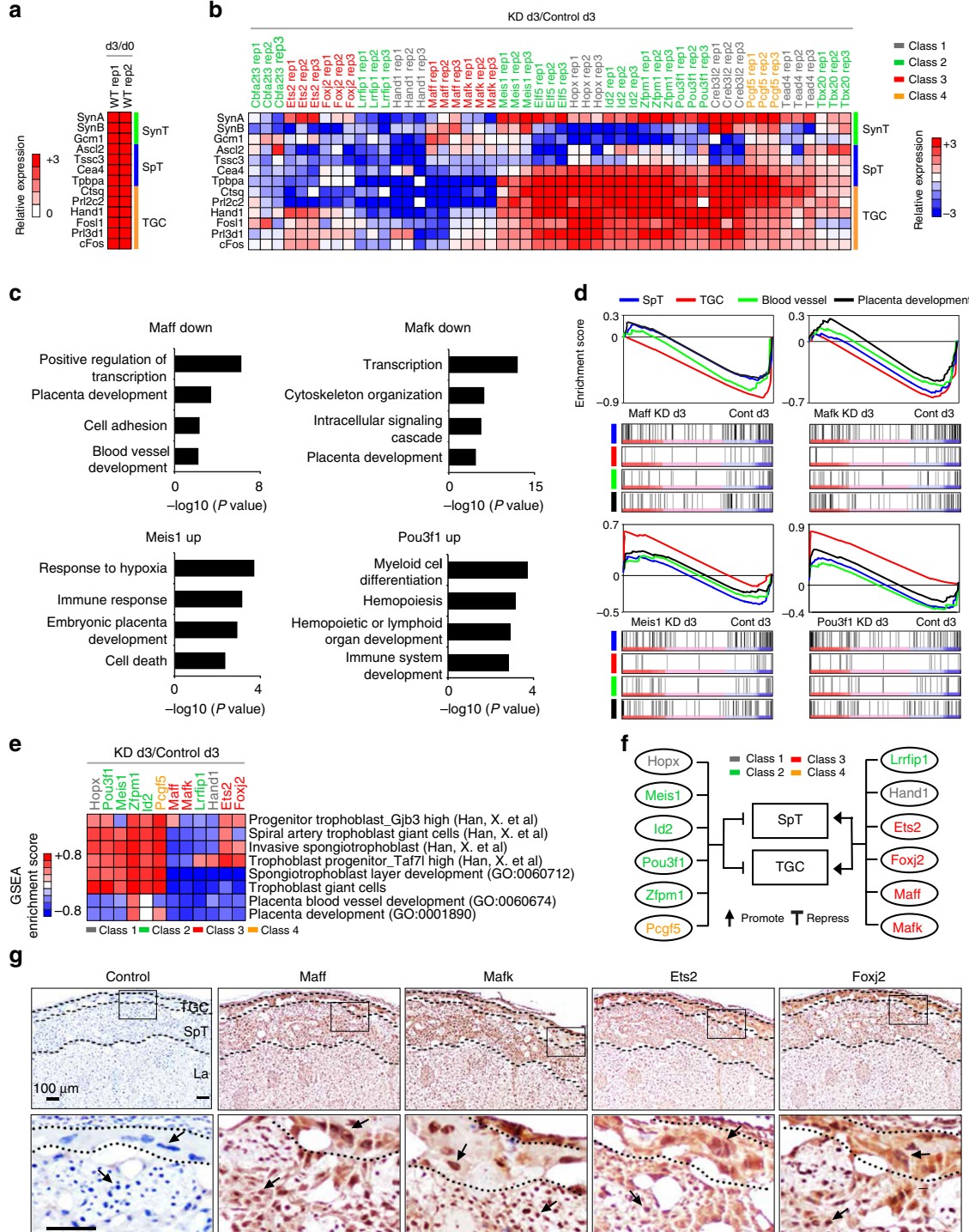

**Fig. 5** Distinct roles of TSC-specific TFs in lineage specification. **a** A heatmap showing relative expression level of each lineage markers upon differentiation (d3). Marker genes of each lineage are shown in *Y*-axis. **b** A heatmap presenting relative expression level of lineage marker genes between a TSC-specific TF knockdown (KD) cells and control cells after 3 days of differentiation. *X*-axis shows TFs that were knocked down. Marker genes of each lineage are shown in *Y*-axis. **c** Enriched GO terms in differentially expressed genes (DEGs) that are directly regulated by TSC-specific TFs. **d** Gene set enrichment analyses (GSEA) of global gene expression profiles, obtained from KD of TFs and control followed by 3 days of differentiation, on diverse groups of genes representing specific placenta-related GO terms. **e** Heatmap showing Enrichment score from GSEA of RNA-seq data obtained from KD of TFs and control followed by 3 days of differentiation on specific placenta-related terms and cell types. Red and blue colors indicate promotion and inhibition of listed terms by KD of a TF, respectively. **f** A schematic diagram presenting regulatory roles of a TF on lineage development of TSCs. Genes and lineages/tissues are shown with oval and rectangular shapes, respectively. Class of each TF is labeled with different colors. **g** Immunohistochemistry of TSC-specific TFs in placenta. TGC, SpT, and La indicate trophoblast giant cell, spongiotrophoblast, and labyrinth, respectively. Each antibody used is shown at the top of the image and control indicates IgG control with hematoxylin staining. Small rectangular box in the upper panel shows an area magnified (bottom panel). An arrow indicates a representative cell in trophoblast lineage and a scale bar depicts 100 μm. Source data underlying Fig. 5a, b are provided as a Source Data file

such as Elf5, Esrrb, Eomes, and Sox2, seem to support TSC self-renewal[13,33–35], whereas Class 3 TFs including Gcm1, Prdm1, Fosl2, Cited2, and Dlx3 seem to orchestrate differentiation toward more specialized placental cell types[37,39,56–58]. Accordingly, the TFs in the Class 3 show dramatic changes in their target gene occupancy upon differentiation, shifting their preference toward marker genes for TGCs and SpTs. Our analysis demonstrates class-specific roles of multiple factors during differentiation, highlighting the complexity of the TE-specific gene expression regulation (Fig. 5e, g). Further characterization of the untested TFs in each class during TE lineage development toward specialized cell types will greatly enhance our understanding of normal and abnormal placentation.

## Methods

**Cell culture/differentiation**. Mouse J1 ES cells (ESCs) were cultivated in Dulbecco's modified Eagle's medium (DMEM) supplemented with 18% fetal bovine serum (FBS), 2 mM L-glutamine, 100 mM nonessential amino acid, nucleoside mix (Fisher Scientific), 100 μM β-mercaptoethanol (Sigma-Aldrich), 1000 U/ml recombinant mouse LIF (Gemini Bio-Products), and 50 U/ml penicillin/streptomycin (Fisher Scientific). Mouse TSCs derived from blastocyst (TS$_{3.5}$) were gifted from Dr. Janet Rossant laboratory. TSCs were maintained in TSC culture media (mixture of mouse embryonic fibroblast (MEF)-conditioned TS medium with TS basal medium at a ratio of 7:3) supplemented with 25 ng/ml Fgf4 and 1 μg/ml of heparin. The TS basal medium is composed of RPMI (Roswell Park Memorial Institute) 1640 medium supplemented with 20% FBS, 100 μM β-mercaptoethanol, 2 mM L-glutamine, 1 mM sodium pyruvate, 50 U/ml penicillin/streptomycin. MEF-conditioned medium was composed of TS medium conditioned by MEF cells. Mitomycin-treated MEF cells were cultivated in TS basal medium for 3 days, and then the medium was collected. 293T cells were cultured in DMEM supplemented with 10% FBS and 50 U/ml of penicillin/streptomycin with 2 mM L-glutamine (Life Technologies). All cells were incubated at 37 °C and 5% CO$_2$. To initiate the differentiation of TSCs, TSCs were washed three times with TS basal medium to remove residual Fgf4 and heparin and then grown in TS basal medium.

**Immunohistochemistry**. For IHC, paraffin tissue sections (ZYAGEN, MP-413-15-C57) of commercial C57 mouse placenta taken at E15 were deparaffinized in xylene (2 times for 5 min). Deparaffinized tissue sections were rehydrated by soaking them in a series of ethanol solution with decreasing concentrations (100% ethanol 2 times for 5 min, 90% ethanol for 3 min, 80% ethanol for 3 min, 70% ethanol for 3 min) and distilled water. Antigens were retrieved from the tissue section slides in pre-heated citrate-EDTA buffer solution (10 mM citric acid, 2 mM EDTA, and 0.05% tween 20, pH 6.2). After heating the slides in water bath for 30 min at 99 °C, the slides were cooled down for 20 min at room temperature (RT) and washed in phosphate-buffered saline (PBS). Before primary antibody treatment, the slides were treated with 0.3% H$_2$O$_2$ in PBS for 10 min at RT and incubated in blocking solution (10% horse serum in PBS) for 30 min at RT. Primary antibody solution prepared in 10% horse serum in PBS with dilution ratio of 1 to 500 was treated in the tissue section slides for 30 min at RT followed by rinsing with PBS for 5 min. The tissue section slides were incubated with biotinylated anti-rabbit or anti-goat IgG antibodies for 30 min at RT. After washing the slides with PBS for 5 min, they were incubated with a VECTASTAIN® Elite® ABC HRP Kit (Vector Laboratories, PK-6100) for 30 min at RT. The tissue section slides were rinsed in PBS for 5 min and developed using the DAB Peroxidase (HRP) Substrate Kit (Vector Laboratories, SK-4100) for 50 s–2 min until the antigen was visibly detected. All sections were counterstained with Harris modified hematoxylin for 3 s. After washing the sections in flowing tap water, they were dehydrated in a graded ethanol series, immersed in xylene, and mounted with VectaMount® Permanent Mounting Medium (Vector Laboratories, H-5000). The slides were examined and photographed using a Nikon Eclipse Ni Compound Light Microscope (Nikon) equipped with a Nikon DS-Ri2 color camera.

**Virus preparation and infections**. shRNA clones targeting each of TFs were purchased from Sigma-Aldrich (Supplementary Data 11). Lentiviruses expressing a specific shRNA were generated by transfecting 6 μg of pLKO vector containing a specific shRNA against a gene of interest, 4 μg of Δ8.9, and 2 μg of VSVG into 9 × 10$^6$ 293T cells. After 12 h, the medium was replaced with the TSC culture medium and then incubated at 37 °C for 36 h. Viral supernatant was collected, filtered with a syringe filter (0.45 μm), and used to infect TSCs. To knock down each factor, 1 × 10$^5$ TSCs in each well of a 12-well plate were infected with viral particles. Infected cells were cultured for 24 h, and the medium was replaced by fresh TSC culture medium with puromycin (1 μg/ml) for the selection of infected cells. For gene expression analysis, cells were harvested after 3 days of infection. For differentiation of TSCs upon KD of each factor, the culture medium was replaced with TS basal medium without Fgf4 and heparin after 3 days of infection.

**Quantitative gene expression analysis**. Total RNAs were extracted from cultivated cells using the RNeasy Plus Mini Kit (Qiagen, 74134). cDNAs were synthesized from 500 ng of total RNAs using qScript cDNA supermix (VWR, 101414-108). To measure gene expression, reverse transcriptase–qPCRs were performed with 1 μl of 20× diluted cDNA for each reaction using PerfeCTa SYBR Green FastMix (VWR, 101414-278) and 250 nM of each primer. PCR primer sets for investigating gene expression were designed to amplify the junction between two exons using a web-based primer design program, Primer3 (http://bioinfo.ut.ee/primer3/). All primer sequences are listed in Supplementary Data 11. CT values of samples and controls were normalized against Gapdh as a loading control, and then relative gene expression was calculated as fold enrichment using the 2$^{-\Delta\Delta CT}$ method. Relative expression of tested genes was calculated from at least duplicate reactions.

**Antibody**. All antibodies used in experiments are listed in Supplementary Data 10.

**Antibody validation**. In order to validate the quality of antibody that we used in ChIP-seq experiments, first, we performed motif analysis from ChIP-seq data to see whether the motif specific to a TF is enriched. We extracted ±100 bp length of sequence from the center of all ChIP-seq peaks and utilized MEME suite[59] for motif search. Among 32 antibodies utilized for our ChIP-seq experiments, motif analysis of 21 TFs (Arid3a, Creb3l2, Ctcf, Elf5, Eomes, Ets2, cFos, Hic2, Irf2, Maff, Mafk, Mef2d, Meis1, Pou3f1, Tead4, Tfap2c, Bhlhe40, Cbfa2t3, Foxj2, Id2, and Tbx20) showed either their own canonical motif or motifs highly similar in sequence to the canonical motif as enriched in the center area of the peaks (Supplementary Fig. 10a). Zfpm1 (Fog1, friend of Gata) is a known interaction partner of Gata3 and its most enriched motif was that of Gata3, suggesting that Zfpm1-binding sites are genuine (Supplementary Fig. 10a). While some TFs (such as Bbx, Fbxo21, Hopx, Lrrfip1, Pcgf5, and Smad6) do not show motif enrichments, the quality of these antibodies were validated by western blotting (WB) or IP-WB (Supplementary Fig. 10b). Four antibodies (H3K4me1, H3K27ac, Med12, and p300) were validated previously (Supplementary Data 10). In sum, for almost all TFs we performed ChIP-seq for, we provide evidence satisfying at least one out of three criteria: reference, motif enrichment, and WB (Supplementary Fig. 10a, b and Supplementary Data 10).

**RNA-seq and data process**. Global gene expression profiles using RNA-seq were performed in both ESCs and TSCs. Total RNAs were isolated using the RNeasy Plus Mini Kit. One μg of total RNA was used to generate RNA-seq library using the NEBNext Ultra RNA Library Prep Kit (NEB, E7530L) according to the manufacturer's instruction. Briefly, mRNAs were isolated from total RNAs with the Magnetic mRNA Isolation Kit, fragmented, and primed. First-strand cDNAs were synthesized following by second-strand synthesis. The ends of purified double-strand cDNAs were repaired, ligated with a barcoded adaptor, and amplified by PCR. Purified final PCR product was sequenced using Illumina NextSeq 500 machine. The reads were aligned to the mouse transcriptome (mm9) using salmon (v0.13.0)[60]. Expression levels of each gene were calculated using R library tximport[61] as transcripts per million and DEGs were defined using P value cut-off of 0.05 (Z-statistic).

**Chromatin immunoprecipitation coupled with next-generation sequencing**. ChIP reactions were performed as previously[45]. Cells were cross-linked with 1% formaldehyde for 7 min at RT, and then formaldehyde was quenched by adding glycine (final 125 mM) for 5 min. After washing cells with PBS two times, fixed cell pellets were resuspended in ChIP dilution buffer and sonicated using a Bioruptor (Diagenode) with a setting of 30 s on and 1 min off for 10 min (3 times). Sheared chromatins containing DNA fragments with an average size of 300 bp were used for IP using 10 μg of a native antibody against each factor. Enriched ChIP samples were used for the generation of sequencing libraries using an NEB ChIP-seq library Preparation Kit (NEB, E6240L) following the manufacturer's instruction. ChIP-seq libraries were sequenced using Illumina NextSeq 500 machine.

**Assay for transposase-accessible chromatin using sequencing**. ATAC assays were conducted as previously described[62]. Briefly, approximately 50,000 cells were incubated with transposition reaction mix for 30 min followed by 18 cycles of PCR. Approximately 250 bp of ATAC samples were isolated using the E-gel Size Select Kit. The final product was sequenced using Illumina NextSeq 500 machine.

**Data processing and peak calling of ATAC-seq and ChIP-seq**. For most of the generated data, at least two independent ChIP-seq and ATAC-seq were performed to identify the binding sites of TFs and open loci of chromatin, respectively. In all, 50 or 75 bp reads from ChIP-seq or ATAC-seq were mapped onto the mouse genome assembly (mm9) using bowtie2[63] and filtered out non-uniquely mapped reads. We combined the mapped reads from the biological replicates for each unique factor and then performed peak calling using the model-based analysis for ChIP-seq (MACS2) peak caller (v2.1.1) with default parameters[64]. Further score filtering was applied to remove weak peaks by visual inspection. In addition, mouse genome mm9 repeat-mask file was downloaded from UCSC table browser

(https://genome.ucsc.edu/cgi-bin/hgTables). Peaks found in simple redundant regions of the genome were further filtered out.

**Moving average analysis**. To investigate the effects of the number of enhancers on gene expression, enhancers (p300-binding sites) were mapped to the region surrounding 20 Kb upstream of the TSS as well as the gene body of all RefSeq genes from the RefFlat file and assigned to the gene. Moving average with a window size of 100 was used to average the number of enhancers that are associated with a gene.

**Identification of SEs**. SEs were defined using the ROSE program downloaded from the website of the Young laboratory (http://younglab.wi.mit.edu/super_enhancer_code.html). We first defined the binding sites of p300 using MACS peak caller, and then the defined peaks were transformed into gff files to meet the criteria of input files of the ROSE program. We run the ROSE with stitching distance option of 12.5 Kb and TSS exclusion zone size option of 2.5 Kb.

**Comparison of SE-associated gene expression with PE genes**. To investigate whether SE-associated genes are abnormally expressed in PE, we downloaded five different PE expression data sets from GEO (GSE30186, GSE25906, GSE43942, GSE44711, and GSE75010) (https://www.ncbi.nlm.nih.gov/geo/). Top 10% and bottom 10% of genes were determined by comparing gene expression between PE and normal samples. If human orthologs of mouse SE-associated genes are detected at least three times among these gene sets, the orthologs were finally selected and provided in Supplementary Data 8.

**Motif analysis**. To analyze the enriched motifs in the peaks from ChIP-seq data, we extracted sequences covering 200 bp centered on each peak. Web-based motif analysis program MEME-ChIP (http://meme-suite.org/index.html) was utilized to identify enriched motifs.

**GO analysis**. Enriched GO terms were investigated using GREAT (v3.0.0)[65] with species assembly of mm9, background regions of whole genomes, and association rule setting of default for a set of loci from ChIP-seq data and DAVID (v6.7)[66] for a set of genes.

**ChIP signal profiling around the center of peaks**. A region of ±3 Kb from the center of peaks was binned (100 bp), and all reads from ChIP-seq were mapped into each bin. Each bin score was calculated by summing up the number of reads assigned into each bin and then normalized with the total sequencing depth. Average bin scores were plotted to generate averaged read density around peaks.

**Mapping peaks to gene features**. To identify distribution of the binding sites of each TF across the genome, TF-binding sites were mapped to the region surrounding 20 Kb upstream and 2 Kb downstream of the TSS of all RefSeq genes from the RefFat file downloaded from UCSC genome browser. To assign one binding site to one genomic feature, we used the following hierarchy: promoter > upstream > intron >exon > intergenic regions. A promoter was defined as a region within ±2 Kb from the TSS, and an upstream element was defined as a region between 2 and 20 Kb upstream from the TSS. Binding sites without being mapped to promoter, upstream, intron, or exon were considered as intergenic target loci.

**Overlap and correlation analyses**. Overlapping binding sites among ChIP-seq data were identified using a moving window across the mouse genome. If the centers of peaks from different ChIP-seq data were discovered within a 500-bp window, we considered them as overlapping peaks. To generate correlation map of the binding sites among different TFs, peak calling followed by an overlap analysis identified the common binding sites of TFs. Score 0 and 1 were assigned to unique and overlapped binding sites of two TFs, respectively. A paired-wise Pearson correlation coefficient between the binding sites of two TFs was calculated for each pair of TFs. Clustering analysis and visualization of the data were done by Cluster 3.0[67] and Java Treeview[68], respectively.

**Network construction and visualization**. To assign a binding site to a gene, TF-binding sites were mapped to the region surrounding 20 Kb upstream and downstream of the TSS as well as the gene body of all RefSeq genes from the RefFlat file. If a binding site was not found within ±20 Kb including the gene body, the nearest gene was assigned as the target of the binding site. TF and target pairs were visualized using Cytoscape (http:// www.cytoscape.org).

**Data used for analyses**. All sequencing data generated or utilized in this study are listed in Supplementary Data 11.

## Data availability
All sequencing data including ChIP-seq, RNA-seq, and ATAC-seq data that support the findings of this study have been deposited in NCBI GEO with the accession codes GSE110950. All other relevant data supporting the key findings of this study are available

within the article and its Supplementary Information files or from the corresponding author upon reasonable request. The source data underlying Fig. 5a, b and Supplementary Figs. 6g and 10b are provided as a Source Data file. A reporting summary for this article is available as a Supplementary Information file.

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

## Acknowledgements

We thank all the members of the Kim laboratory for their help and the Genome Sequencing and Analysis Facility (GSAF) at UT-Austin for NGS. The project is partially supported by R01GM112722 from the National Institute of General Medical Sciences and J.K. holds a 2017 Preterm Birth Research Grant from the Burroughs Wellcome Fund.

## Author contributions

B.-K.L. and J.K. conceptualized the study. B.-K.L., C.R., W.S., Y.j.J., M.K. and J.L. performed experiments. B.-K.L., S.B. and J.K. performed data analyses. B.-K.L., L.L. and J.K. wrote the manuscript.

## Competing interests

The authors declare no competing interests.
