## [Peer Review File · Nature Communications]

Reviewers' comments:

Reviewer #1 (Remarks to the Author):

Lee et al have performed an extensive profiling effort of the TSC regulatory landscape, including characterizing 'super-enhancers' and performing ChIP-seq for a large group of transcription factors, some of which had not been previously implicated in trophoblast development. The involvement of a subset of these TFs in TSC differentiation was analysed by depletion experiments, revealing differential roles in stemness/differentiation.

There are a number of valuable insights gained from this work that forward our knowledge of TSC biology, which continues to lag far behind that of ESCs, despite the vital importance of the placenta. Additionally, the work provides a large body of data that are useful to the community, as well as novel molecular targets for further investigation in vivo. I have the following comments/suggestions:

1. A number of SE-associated TFs are associated with TSC differentiation and some indeed play a role as judged by the knockdown experiments. However, it wasn't clear to me whether the TSC SEs to which they are meant to bind (and possibly non-SE binding sites) are maintained during differentiation, such that they are perhaps in a poised conformation in TSCs and/or play dual roles depending on which TFs are bound. Could the authors please comment and/or provide additional analyses to clarify this point?

2. If I understand correctly, regulatory regions were assigned to genes if they were either within 20kb upstream of the TSS or within the gene body. This strikes me as fairly odd criteria, given that the region downstream of the TSS that is considered will vary depending on gene length. Seems to me it would make more sense to just apply the same distance threshold for binding events upstream or downstream of the TSS, given that enhancer action does not depend on their position relative to gene bodies. Did the authors also consider using the recently published promoter capture Hi-C data in TSCs (PMID 30305613) to assign regulatory elements to genes?

3. Could the authors provide additional support for specific regulation of SynT formation by using additional markers? Currently Fig. 5b relies solely on Gcm1 expression. SynA and SynB would be possible additional markers.

4. Could the authors provide quality metrics for all of their ChIP-seq data? This is hard to judge from Fig. 3a, for example, where some of the tracks (e.g., Pcgf5, Tbx20) give the impression that these are

of poorer quality. Of course the answer could be that simply there are no peaks in this region, but there's no way to distinguish between the two. One alternative could also be to provide links to bigwig tracks so that the data can at least be visually inspected by reviewers (and readers).

5. The methods section could do with more details in a number of places. For example:

a) "Chromatin immunoprecipitation (ChIP) reactions were performed as previously". There is a reference missing here without having provided full details of the protocol subsequently.

b) "Briefly, tagmentation reaction was performed with 50 ng of target DNA". Surely for ATAC-seq this was not performed on purified DNA but on chromatin. Please clarify and provide details about cell lysis and downstream steps.

c) After alignment with bowtie2 were multi-hit reads filtered out to keep only uniquely mapped reads? This is not specified and the default settings in bowtie2 will report multi-hit reads

d) Please specify the settings used in GREAT.

6. Axis labels are missing in Figs. 3a, 4a, 5c, 5d, Ext figs. 1a, 1g, 2a, 5e, 6c, 6h. Also labels missing in heatmap scales in Fig. 4c, Ext figs. 3c, 6b, 6e. I might have missed some.

Reviewer #2 (Remarks to the Author):

For clarity I will answer the specific questions defined in the Guide for Referees, followed by additional comments/queries:

Q1. What are the major claims of the paper?

The present manuscript presents a large-scale functional genomics study focusing on transcriptional networks involved in mouse trophoblast stem cell (TSC) self-renewal and differentiation. First the authors use p300 ChIP-seq to identify enhancers and super-enhancers (SEs) functioning in TSCs and show that they exist proximal to potentially interesting genes including transcription factors (TFs) of which some are already established to have important functions in the trophoctoderm lineage. The authors then categorize TFs into four classes based on their expression profiles during TSC differentiation and select a large number of TFs across the classes for ChIP-seq. They show that

many TF localize to the same SEs, and that TF binding is largely driven by context (i.e. stage of TSC differentiation) rather than TF identity. They then show that shRNA knockdown (KD) of studied TFs is consistent with known phenotypes, and that KD of novel TFs affects TSC differentiation consistent with the temporal expression profiles of the TFs.

In my view the two major contributions that the manuscript could make are by revealing new TFs with important roles in TSC differentiation (mainly Maff, Mafk, Meis1 and Pou3f1), and providing a resource of functional genomics data to understanding the mechanistic basis for their loss-of-function phenotypes.

Q2. Are the claims novel? If not, please identify the major papers that compromise novelty.

The claims are largely novel, though some of the data are somewhat redundant with earlier functional genomics approaches to identify enhancer use in TSCs including Rugg-Gunn et al. (2010) PNAS, 107:10783-10790; Chuong et al. (2013) Nat Genet, 45:325-9; Nelson et al. (2017) Sci Rep, 7:6793. I would argue that while the present manuscript focuses on the subset of super-enhancers (which are likely to be represented in the existing published data, though not explicitly identified as super-enhancers), that mention of the preceding literature may be appropriate. Though perhaps less directly relevant, Tuteja et al., (2016) Placenta, 37:45-55 also attempted to identify enhancers in embryonic placenta with some consideration of super-enhancers.

The TF CHIP-seq data is novel (existing Tfp2c data at a different stage notwithstanding) and may serve as a useful resource to other groups wishing to study these TFs. However, the quality and validity of the data has not been established (see below).

Identification of novel TFs influencing TSC differentiation is novel and potentially important, though how these TFs might influence TSC differentiation through regulation of their target genes is not comprehensively explored/discussed in the manuscript.

Q3. Will the paper be of interest to others in the field?

As previously stated, I think the paper has the potential to be a useful resource. The identification of SEs in TSCs is novel (to the best of my knowledge), but not wholly surprising. I also feel that since regular enhancers outside of super-enhancers also have significant function that the focus on SEs doesn't seem completely justified. Though the boxplot in Extended Data Fig. 2d shows that SEs are

associated with more highly expressed genes the difference is not compelling enough to suggest that regular enhancers are not also very important.

Q4. Will the paper influence thinking in the field?

In my view the biggest effect this paper may have is in inspiring researchers to investigate the novel transcription factors shown to influence TSC differentiation. It may not profoundly change how we understand trophoblast gene regulation, though this paper does now specifically name new likely regulators. The paper quite nicely suggests that there is a highly integrated transcriptional network with many TFs regulating themselves and each other (Extended Data Fig. 5d). But the authors do not specifically test the effects of removing individual TFs on the expression of their putative targets from ChIP-seq, or on the TF transcriptional network in general. Since not all TF DNA-binding events are likely to be functionally important it is not fully established to what extent the inferred feedback, feed-forward and autoregulatory mechanisms contribute.

Q5. Are the claims convincing? If not, what further evidence is needed?

This manuscript includes and relies on ChIP-seq data for a large number of TFs, however, the quality and validity of these data is entirely dependent on the specificity of the antibodies. No information is provided on this so it is not possible to determine whether the data are publishable and conclusions accurate. One form of validation that is required, though is not individually sufficient, is a demonstration that known binding sequences for sequence-specific binding factors are enriched under the ChIP-seq peaks. Such analyses are generally not present, though they were performed for the super-enhancers they defined so could easily extend this to all ChIP-seq peaks. It would be acceptable to provide references to other papers where antibodies have been appropriately validated in ChIP experiments. Otherwise I would argue that additional immunoprecipitation and Western blot experiments for antibodies are required to prove specificity.

In multiple places functional and anatomical annotation analysis was performed to infer the function of TFs. However, given that chromatin accessibility tends to reflect cell identity and function, TFs that bind accessible chromatin will always yield results reflecting this regardless of whether those TFs are important in influencing the expression of the presumed target genes. For such annotation enrichment analyses to reflect function I believe that they have to consider the intersect of TF ChIP-seq targets and gene expression changes on TF KD or deletion.

Q6. Are there other experiments that would strengthen the paper further? How much would they improve it, and how difficult are they likely to be?

Questions surrounding antibody specificity need to be addressed, as mentioned above. What experiments are appropriate depend on what data remains after ChIP-seq quality control/validation.

I would like to see an analysis of the intersect between ChIP-seq data and KD RNA-seq data for the novel TFs in Fig. 5 and some discussion of how this might explain the differentiation phenotypes. I would also ideally like to see a more comprehensive analysis of TF KD across the whole panel of TFs to establish the importance and validity of the conclusions derived from the ChIP-seq data.

To establish whether Maff, Mafk, Meis1 and Pou3f1 KD results are likely to reflect in vivo reality rather than being an in vitro TSC phenomenon I would also like to see in vivo expression patterns and a discussion of how the new in vivo results relate to the existing in vitro data.

These experiments/analyses are unlikely to be particularly technically challenging but may be demanding in terms of workforce and expense.

Q7. Are the claims appropriately discussed in the context of previous literature?

Largely, yes. Though it may be appropriate to cite the preexisting literature on TSC enhancer usage and the chromatin landscape as mentioned above. Any new analyses, however, are likely to require reference to more of the previous literature.

Q8. If the manuscript is unacceptable in its present form, does the study seem sufficiently promising that the authors should be encouraged to consider a resubmission in the future?

If the authors can address the above then I think a resubmission would be worthwhile, though this seems challenging in a reasonable timeframe.

Further comments:

1. Use of yellow fonts in figures makes the text too hard to read.

2. The use of the TSC column in the heatmaps in Figure 2 is very helpful but also slightly confusing since it is also red. Would using a different color scheme to the timecourse heatmap be clearer?
3. To emphasize or refute the importance of SEs vs REs it would be informative to see the analysis for Extended Data Fig. 2i for REs.
4. It would be informative to know what fraction of TF binding events do not coincide with p300. Could the Extended Data Fig. 3f heatmaps be extended to include peaks not at p300 sites?
5. How variable is the distribution of TF binding sites for different TFs? Is Extended Data Fig. 3a meaningful?
6. In the legend for Extended Data Fig. 3d it should be explained how “proximal” and “distal” are defined.
7. There is a misalignment in the Extended Data Fig. 4b axis label.
8. Tables of read numbers and mapping statistics should be provided to indicate ChIP-seq data quality.
9. Page 5, paragraph 1 – was 500µg of total RNA really used or is it a typographical error? 500ng perhaps? Also referring to volumes of cDNA is not informative without the concentration being clear.
10. Page 6, para 3 – what version of MACS was used and with what parameters?
11. Page 6, para 4 – can a rationale for the window parameters e.g. 20kb upstream be provided?
12. Page 6, para 5 – I think it should read “...website of the Young lab...”.
13. Page 7, para 1 – can taking the top and bottom 10% of genes from each dataset rather than making a judgment based on statistical significance be justified?
14. Page 7, para 3 – what versions and parameters were used for GREAT and DAVID?

Reviewer #3 (Remarks to the Author):

Understanding the regulatory networks that control trophoblast development and its differentiation to the different lineages that conform the mature placenta is of great interest but, as the authors point out, has been less extensively studied than other cell populations of the early mammalian embryos that contribute to the embryo proper. In this regard, the work presented here

by Lee and colleagues is of clear interest and could help to complement our knowledge of this process.

In order to do so, the authors carry out an extensive genomic characterization of trophoblast stem cells (TSCs) used as a proxy for trophectoderm lineage development. Using a combination of ChIP-seq, RNA-seq and ATAC-seq on multipotent and differentiated TSCs, together with knock-down of candidate genes, the authors describe some novel transcriptional regulators that could be involved in TSC maintenance and differentiation.

Overall, this data-rich manuscript provides a very useful resource for the community that will surely benefit from the different genomic analysis performed on TSCs. However, despite the impressive amount of data and analysis, this work falls short of providing a convincing physiological context for the novel findings obtained in a culture system. Major issues in this regard are discussed below.

1. The authors use the characterization of super-enhancers in TSCs to identify novel genes involved in TE and placenta biology. Although the description of super-enhancers in another cell type is interesting, in this work there is little novel insight into the nature and function of these regulatory elements. For sure this is not the aim of the work, but the value of using super-enhancers to identify novel genes important in TSCs is not properly justified. Couldn't the authors have used regular enhancers equally? Or simply differential gene expression? The fact that the authors link more than one thousand genes to TSC super-enhancers would argue that this approach is not that useful to pinpoint key regulators.

Despite the strong emphasis on super-enhancers, we do not learn much about their function or role in gene regulation more than correlative analysis. The authors claim cooperativity and synergistic effects of transcriptional inputs on these regulatory elements, but this is based only from the presence of putative TF binding sites on the super-enhancers and in some cases co-occupancy as determined by ChIP-seq. Taking into account that the size of super-enhancers ranges from 1 up to 135 kb, presence of a consensus binding site sequence cannot be taken as evidence of regulation by a given factor. No functional assays are provided to prove that these elements are having a role in TSCs. Without this sort of evidence, all analysis shown are merely correlative and conclusions based on super-enhancer predictions should be toned down.

2. The authors compare enhancer predictions and several TF occupancy patterns between multipotent and differentiated TSCs. Here, little detail is provided regarding the differentiation protocol and the resulting cell types. Differentiation was induced by removing Fgf4 and heparin, what has been described to result mainly in differentiation of trophoblast giant cells but not other

trophoblast-derived cell types (reviewed in Latos & Hemberger, 2016, *Development* 143, 3650-60). Therefore, without further characterization, we do not really know what cell types are obtained. No time course or marker expression is shown in order to more carefully understand the system used to analyze differentiating TSCs.

This problem is most clearly shown in Fig. 5a, where apparently after 3 days, markers for all different cell types examined are present simultaneously. Furthermore, using a single marker for a specific subtype, as is the case of *Gcm1* for SynT, is rather weak to drive any definitive conclusions. This panel is used then to describe the effect of the knock-down of a number of newly identified transcription factors that could be involved in TSC differentiation (Fig 5b). According to the description of this experiment in Methods, RT-qPCR was performed at least in duplicates, and changes are shown as fold changes. Although the results seem robust, no statistical analysis is provided for this experiment. RNA-seq for a number of these KD was generated, so more quantitative data should be available. Are the changes depicted in Fig. 5b statistically significant in standard differential gene expression analysis of RNA-seq data for *Maff*, *Mafk*, *Meis1* or *Pou3f1* KDs?

4. The most interesting and central claim of this manuscript is the identification of novel transcription factors with a previously undescribed role in TSCs, what would extend our knowledge of the gene regulatory networks underlying TE and placental development. Starting from more than 150 transcription factors initially identified in their super-enhancer based screen, the authors examine genome-wide binding of 27 of these in TSCs and of a subset in differentiated TSCs. The logic for the selection of these factors is unclear and should be described (antibody availability for ChIP-seq?). Network analysis of this data leads the authors to further select a handful of these for KD experiments, leading to define novel regulatory interactions as depicted in Fig. 5e.

Data shown to support these claims seems at most preliminary, and further insight should be provided. For example, the effect of combinatorial KD of these factors would shed light on their supposed cooperativity and synergistic action. Most importantly, the physiological and *in vivo* relevance of the findings needs to be proven. At least, the description of the expression patterns of a number of these genes during placenta development should be provided to support them having a role in this process. Re-examining mutant phenotypes would also be desirable, although understandably this would be out of the scope of the present manuscript.

Summary of additional/modified data/text in response to the reviewer's comments:

Reviewer	Comment	Response: additional/modified data/text
Reviewer 1	1-1	Response Figure 1
	1-2	Response Table 1* and updated Supplementary Table 7
	1-3	Updated Fig. 5b and modified text in Results section
	1-4	Response Figure 2*
	1-5-a	Reference added in Supplementary Information (page 7)
	1-5-b	Reference added in Supplementary Information (page 7) and text corrected
	1-5-c	See point-by-point response
	1-5-d	Text edited in Supplementary Information (page 9)
	1-6	All missing labels added
Reviewer 2	2-1	See point-by-point response
	2-2	All papers cited in Introduction (page 4) See our response to the points 2-4, 2-5, and 2-6
	2-3	Response Figure 3 and updated Supplementary Table 4
	2-4	Main text edited (page 8) Updated Fig. 5b, 5c and Supplementary Fig. 7a, 7b
	2-5	Response Figures 4 and 5 and Response Table 2 Updated Fig. 5c and Supplementary Fig. 7
	2-6	See our response to the point 2-5 Updated Fig 5c, 5e, 5f, 5g and Supplementary Fig. 7, 8c
	2-7	All papers cited in Introduction (page 4)
	2-8	See point-by-point response
	Minor-1	Modified font color
	Minor-2	Modified Fig. 2 color schemes
	Minor-3	Response Figure 3
	Minor-4	Supplementary Fig. 3e
	Minor-5	Supplementary Fig. 3a, 3c
	Minor-6	Text edited in Supplementary Information (page 2)
Minor-7	Misalignment corrected in Supplementary Fig. 4b	
Minor-8	See point-by-point response	
Minor-9	Text corrected in the Supplementary Information (page 6)	
Minor-10	Text edited in Supplementary Information (page 8)	
Minor-11	See our response to the point 1-2	
Minor-12	Typo corrected in Supplementary Information (page 8)	
Minor-13	See point-by-point response	
Minor-14	Text edited in Supplementary Information (page 9)	
Reviewer 3	3-1-a	Response Figure 3 and updated Supplementary Table 4
	3-1-b	Response Figure 6
	3-2	Response Figure 7
	3-3	Response Figure 8 and updated Fig. 5b
	3-4-a	See point-by-point response
	3-4-b	Updated Fig. 5e, 5f, 5g

*See pages 15 and 16 for Response Tables 1 and 2, respectively.

Overall Response:

We appreciate all reviewers for their insightful comments and constructive suggestions. We thank the reviewers for seeing the potential of our work to be a useful resource and foundation for further studies in the field of trophoblast biology. We also recognize that main criticisms are about the 1) logic of using SE to map putative key regulators, 2) quality control of ChIP data, 3) necessity of testing more factors using KD approach and integrative analyses of RNA-seq and ChIP-seq data, and 4) lack of *in vivo* validation. To improve our manuscript and provide more convincing data, we performed additional experiments and analyses as follows:

1. While acknowledging the importance of REs, we clarify the benefits of using SEs to map putative cell type-specific key regulators.
2. We performed rigorous quality control to validate our ChIP-seq methods by monitoring enriched target DNA motifs, validating antibodies by Western Blot (WB) and/or IP-WB in addition to literature search for prior usage of the antibodies in the current study.
3. To support our existing 5 RNA-seq data sets, we performed an additional 7 RNA-seq experiments upon KD of Ets2, Foxj2, Hand1, Id2, Lrrfp1, Pcgf5, and Zfp1 followed by 3-day differentiation. In total, 12 RNA-seq data sets were analyzed and 11 of them were used to perform integrative analysis with corresponding ChIP-seq data.
4. We examined the *in vivo* expression of multiple novel TSC-specific TFs in mouse placenta by immunohistochemistry (IHC).

Additionally, we corrected all errors that the reviewers pointed out and tried to answer the questions and concerns raised. As the reviewers noted, we believe that the current research will serve as an important resource for the field of trophoblast biology and a useful framework for understanding how TSC-specific TFs contribute to placental lineage specification. The following is a point-by-point response to each comment, which we hope will be clear and comprehensive.

Reviewer #1:

Lee et al have performed an extensive profiling effort of the TSC regulatory landscape, including characterizing 'super-enhancers' and performing ChIP-seq for a large group of transcription factors, some of which had not been previously implicated in trophoblast development. The involvement of a subset of these TFs in TSC differentiation was analysed by depletion experiments, revealing differential roles in stemness/differentiation. There are a number of valuable insights gained from this work that forward our knowledge of TSC biology, which continues to lag far behind that of ESCs, despite the vital importance of the placenta. Additionally, the work provides a large body of data that are useful to the community, as well as novel molecular targets for further investigation *in vivo*. I have the following comments/suggestions:

1. A number of SE-associated TFs are associated with TSC differentiation and some indeed play a role as judged by the knockdown experiments. However, it wasn't clear to me whether the TSC SEs to which they are meant to bind (and possibly non-SE binding sites) are maintained during differentiation, such that they are perhaps in a poised conformation in TSCs and/or play dual roles depending on which TFs are bound. Could the authors please comment and/or provide additional analyses to clarify this point?

Response: We appreciate the reviewer's an intriguing question. One interesting observation we had was that some SE-associated TFs defined in TSCs show even more increased activity upon differentiation of TSCs (Class 3 TFs). We found that these TFs are associated with context-

Response Figure 1. ChIP-seq signal tracks of p300 in trophoblast stem cells (TSCs) and differentiated trophoblast stem cells (dTSCs).

dependent SEs as shown in **Response Figure 1** examples. This suggests that some TFs active in both TSCs and differentiated TSCs may execute transcription by the interaction with their context-dependent target enhancers before and after differentiation of TSCs (**Fig. 4c**). Since the Class 3 TFs are already active in self-renewing TSCs, we do not consider the SEs that are associated with these TFs as poised. However, this class of TFs has not been shown in the study of ESCs as most of the ESC-specific SE-associated TFs are Class 2-like factors (active in ESCs and become inactive upon differentiation). This also emphasizes the benefit of using SEs to predict key regulators as an exclusive approach using expression profiling may not detect this type of regulators.

2. If I understand correctly, regulatory regions were assigned to genes if they were either within 20kb upstream of the TSS or within the gene body. This strikes me as fairly odd criteria, given that the region downstream of the TSS that is considered will vary depending on gene length. Seems to me it would make more sense to just apply the same distance threshold for binding events upstream or downstream of the TSS, given that enhancer action does not depend on their position relative to gene bodies. Did the authors also consider using the recently published promoter capture Hi-C data in TSCs (PMID 30305613) to assign regulatory elements to genes?

Response: We agree with the reviewer's comment and in turn, we analyzed the sequence 20 Kb upstream and downstream of the TSSs, including the gene body, to find out the number of associated enhancers shown in **Fig. 1c**. Since our description in the original manuscript was wrong, we corrected the mistake accordingly. While some researchers prefer to assign one binding site to a nearest gene regardless of directions¹⁻³, some use 10 Kb upstream and 3 Kb downstream distance from TSS criteria⁴, all of which are arbitrary.

We strongly agree with the reviewer that the better way of assigning enhancers to genes may be considering topologically associated domains (TADs) in order to circumvent false assignments of the binding sites to genes. However, one caveat of this way of assignment is that Hi-C data typically has resolution ranging from 40 Kb and 1 Mb, reducing the precision of regulatory element assignment when many genes are clustered together in less than 40 Kb distance^{5,6}. Regardless of this issue, we downloaded the Hi-C data from the paper the reviewer suggested⁷ and assigned the enhancers to genes using the looping information. When we assigned enhancers to target genes considering the promoter and distal interactions, we found that approximately 15 to 61 % of binding sites (among 30 TFs) are assigned to the genes (**Response Table 1**). Due to the lack of multiple Hi-C data sets to compare, we are not sure whether this is attributed to biology or the quality of this particular Hi-C data set. Therefore, while we would like to use our own criteria, we provided raw data as well as the binding sites of each TF tested to allow other researchers to apply their own criteria. We also provide the targets of TFs based on the criteria of the promoter and distal interactions mapped by the Hi-C method (updated **Supplementary Table 7**). If a binding site of a TF is associated with a promoter of a gene via looping defined by the Hi-C data, we called the gene as a target of a TF.

3. Could the authors provide additional support for specific regulation of SynT formation by using additional markers? Currently Fig. 5b relies solely on Gcm1 expression. SynA and SynB would be possible additional markers.

Response (also for the Reviewer's point 3-3 in part): Thank you for this comment. As suggested, we have performed qPCR for SynA and SynB. Gcm1 and SynB are markers of syncytiotrophoblast layer II (SynT-II) and SynA is a marker of layer I (SynT-I)⁸. We found that SynB and Gcm1 show a similar expression pattern when TSC-specific TFs are depleted, while SynA shows an opposite expression pattern. This suggest that the multiple TFs (Elf5, Hopx, Id2, Pou3f1, and Zfp1) promotes SynT-II differentiation while inhibits SynT-I differentiation. As we observe the differential expression patterns of SynT marker genes, we updated our **Fig. 5b** and discussed it in the main text, accordingly.

4. Could the authors provide quality metrics for all of their ChIP-seq data? This is hard to judge from Fig. 3a, for example, where some of the tracks (e.g., Pcgf5, Tbx20) give the impression that these are of poorer quality. Of course the answer could be that simply there are no peaks in this region, but there's no way to distinguish between the two. One alternative could also be to provide links to bigwig tracks so that the data can at least be visually inspected by reviewers (and readers).

Response: The total target peak numbers of these two factors are relatively smaller than other TFs as shown in the page 5 in the **Reporting Summary** file, but there are many strong peaks as shown in the **Response Figure 2**, suggesting that the binding sites of the factors are likely to be genuine. As requested, we also provided the link of signal tracks for all ChIP-seq data (http://genome.ucsc.edu/s/bumlee92/TSC_all_track). Furthermore, we performed quality control of our ChIP-seq data by using motif analysis and antibody validation. Please also see our response to the reviewer's comment 2-5 (Q5).

5. The methods section could do with more details in a number of places. For example:

a) "Chromatin immunoprecipitation (ChIP) reactions were performed as previously". There is a reference missing here without having provided full details of the protocol subsequently.

Response: We are sorry for the missing reference. We added the reference at page 7 in the Supplementary Information.

b) "Briefly, tagmentation reaction was performed with 50 ng of target DNA". Surely for ATAC-seq this was not performed on purified DNA but on chromatin. Please clarify and provide details about cell lysis and downstream steps.

Response: Thank you for pointing this out. We replaced the method with adding a reference at page 7 in the Supplementary Information and now it reads as "ATAC assays were conducted as previously⁹. Briefly, approximately 50,000 cells were incubated with transposition reaction mix for 30 min followed by 18 cycles of PCR reactions. ~250 bp of ATAC samples were isolated using E-gel size select kit. The final product was sequenced using Illumina NextSeq 500 machine."

c) After alignment with bowtie2 were multi-hit reads filtered out to keep only uniquely mapped reads? This it not specified and the default settings in bowtie2 will report multi-hit reads.

Response: Bowtie2 actually filters multi-hit reads automatically based on score. Bowtie2 manual explains that "Default mode: search for multiple alignments, report the best one. By default, bowtie2 searches for distinct, valid alignments for each read. When it finds a valid alignment, it continues looking for alignments that are nearly as good or better. The best alignment found is reported (randomly selected from among the best if tied)". This indicates that only one randomly selected alignment is reported for a read when multi-hit reads are detected during alignment.

d) Please specify the settings used in GREAT.

Response: We specified the settings at page 9 in the Supplementary Information. Now it reads as "Enriched GO terms were investigated using GREAT (v3.0.0)¹⁰ with species assembly of mm9, background regions of whole genomes, and association rule setting of default for a set of loci from ChIP-seq"

6. Axis labels are missing in Figs. 3a, 4a, 5c, 5d, Ext figs. 1a, 1g, 2a, 5e, 6c, 6h. Also labels missing in heatmap scales in Fig. 4c, Ext figs. 3c, 6b, 6e. I might have missed some.

Response: We are sorry for the missing labels. Now, we have added all the missing axis labels in this revised version of manuscript and figures.

Reviewer #2:

For clarity I will answer the specific questions defined in the Guide for Referees, followed by additional comments/queries:

Q1. What are the major claims of the paper?

The present manuscript presents a large-scale functional genomics study focusing on transcriptional networks involved in mouse trophoblast stem cell (TSC) self-renewal and differentiation. First the authors use p300 ChIP-seq to identify enhancers and super-enhancers (SEs) functioning in TSCs and show that they exist proximal to potentially interesting genes including transcription factors (TFs) of which some are already established to have important functions in the trophectoderm lineage. The authors then categorize TFs into four classes based on their expression profiles during TSC differentiation and select a large number of TFs across the classes for ChIP-seq. They show that many TFs localize to the same SEs, and that TF binding is largely driven by context (i.e. stage of TSC differentiation) rather than TF identity. They then show that shRNA knockdown (KD) of studied TFs is consistent with known phenotypes, and that KD of novel TFs affects TSC differentiation consistent with the temporal expression profiles of the TFs.

In my view the two major contributions that the manuscript could make are by revealing new TFs with important roles in TSC differentiation (mainly Maff, Mafk, Meis1 and Pou3f1), and providing a resource of functional genomics data to understanding the mechanistic basis for their loss-of-function phenotypes.

Response: We thank the reviewer for the critical insights, comments and suggestions to improve our manuscripts. Thanks these comments, we think that the revised manuscript has become more rigorous and convincing.

Q2. Are the claims novel? If not, please identify the major papers that compromise novelty.

The claims are largely novel, though some of the data are somewhat redundant with earlier functional genomics approaches to identify enhancer use in TSCs including Rugg-Gunn et al. (2010) PNAS, 107:10783-10790; Chuong et al. (2013) Nat Genet, 45:325-9; Nelson et al. (2017) Sci Rep, 7:6793. I would argue that while the present manuscript focuses on the subset of super-enhancers (which are likely to be represented in the existing published data, though not explicitly identified as super-enhancers), that mention of the preceding literature may be appropriate. Though perhaps less directly relevant, Tuteja et al., (2016) Placenta, 37:45-55 also attempted to identify enhancers in embryonic placenta with some consideration of super-enhancers.

Response: We appreciate the reviewer's suggestion and cited all above-mentioned works in our revised Introduction at page 4.

The TF ChIP-seq data is novel (existing Tfap2c data at a different stage notwithstanding) and may serve as a useful resource to other groups wishing to study these TFs. However, the quality and validity of the data has not been established (see below).

Response: Please see our response to the point 2-5 (Q5).

Identification of novel TFs influencing TSC differentiation is novel and potentially important, though how these TFs might influence TSC differentiation through regulation of their target genes is not comprehensively explored/discussed in the manuscript.

Response: Please see our response to the points 2-4 (Q4), 2-5 (Q5), and 2-6 (Q6).

Q3. Will the paper be of interest to others in the field?

As previously stated, I think the paper has the potential to be a useful resource. The identification of SEs in TSCs is novel (to the best of my knowledge), but not wholly surprising. I also feel that since regular enhancers outside of super-enhancers also have significant function that the focus on SEs doesn't seem completely justified. Though the boxplot in Extended Data Fig. 2d shows that SEs are associated with more highly expressed genes the difference is not compelling enough to suggest that regular enhancers are not also very important.

Response (also for the Reviewer's points 1-1 and 3-1): We do not intend to claim that REs are not important in controlling cellular identity. We fully acknowledge that they are instrumental in orchestrating cell type-specific gene expression. However, mapping SEs provides an alternative and valuable approach to predict master regulators, and this has been performed in several other cellular contexts^{1,11,12}. As shown in **Supplementary (previously Extended Data) Fig. 2a**, we confirmed that most of known TSC-specific TFs are associated with SEs or multiple enhancers mapped in TSCs (**Supplementary Table 5**). We additionally provide boxplot data from the analysis of RE-associated genes in various tissues (**Response Figure 3**), showing that expression patterns of these genes are not significantly different among multiple tissues. To address reviewers' concern, we toned down the importance of SEs and acknowledged the importance in RE-related factors in modulating cellular identity. We also provide the list of RE-associated genes in updated **Supplementary Table 4** in our revised manuscript. We hope that the revised manuscript will convey our intention more clearly.

Q4. Will the paper influence thinking in the field?

In my view the biggest effect this paper may have is in inspiring researchers to investigate the novel transcription factors shown to influence TSC differentiation. It may not profoundly change how we understand trophoblast gene regulation, though this paper does now specifically name new likely regulators. The paper quite nicely suggests that there is a highly integrated transcriptional network with many TFs regulating themselves and each other (Extended Data Fig. 5d). But the authors do not specifically test the effects of removing individual TFs on the expression of their putative targets from ChIP-seq, or on the TF transcriptional network in general. Since not all TF DNA-binding events are likely to be functionally important it is not fully established to what extent the inferred feedback, feed-forward and autoregulatory mechanisms contribute.

Response: We agree with the reviewer that this is an important issue. While TF occupancy-based approaches have been tremendously useful for understanding of underlying mechanisms of global transcriptional gene regulation¹³⁻¹⁶, many studies showed that the expression of approximately 1~10% of target genes are affected by perturbation of a TF¹⁷⁻²⁰. As this observation is attributed to various possibilities such as wrong target assignment, insufficient KD, functional redundancy and compensation, and functions that are necessary, but not sufficient²¹, we cannot simply ignore the binding sites which do not show gene expression changes upon KD of a TF.

Therefore, we toned down the sentence regarding the regulatory mechanisms mediated by studied TFs. Now it reads as “when assuming regulator binding to a gene implies regulatory control, TSC-specific TRN implies that most of the TSC-specific-TFs in the network are regulated by feed-forward, feedback, and auto-regulatory mechanisms” in page 8 of the revised manuscript.

In addition, by performing 7 additional RNA-seq, we tested the relationship between the target occupancy of a total of 11 TFs and gene expression upon KD of each TF followed by differentiation. First, we intersected targets of differentially expressed genes (DEGs) with our ChIP-seq data and performed GO analysis to understand how and to what extent direct targets of a TSC-specific TF influence differentiation of TSCs. Consistent with our qPCR results (Fig. 5b), depletion of the Class 2 TFs yielded DEGs that mostly showed enrichment in placenta development-related GO terms including placenta development, blood vessel development, or cytoskeleton organization in their down-regulated target genes. Interestingly, up-regulated target genes upon KD of Class 3 TFs were associated with placenta development-related GO terms, such as female pregnancy, blood vessel development, or hematopoiesis (Fig. 5c and Supplementary Fig. 7). These results suggest that TSC-specific TFs directly control the genes involved in placenta development, but their modes of action are Class-specific.

Q5. Are the claims convincing? If not, what further evidence is needed?

This manuscript includes and relies on ChIP-seq data for a large number of TFs, however, the quality and validity of these data is entirely dependent on the specificity of the antibodies. No information is provided on this so it is not possible to determine whether the data are publishable and conclusions accurate. One form of validation that is required, though is not individually sufficient, is a demonstration that known binding sequences for sequence-

specific binding factors are enriched under the ChIP-seq peaks. Such analyses are generally not present, though they were performed for the super-enhancers they defined so could easily extend this to all ChIP-seq peaks. It would be acceptable to provide references to other papers where antibodies have been appropriately validated in ChIP experiments. Otherwise I would argue that additional immunoprecipitation and Western blot experiments for antibodies are required to prove specificity.

Response: We thank you for this critical comment. In order to show the validity of our ChIP-seq data, first, we performed motif analysis as suggested. We extracted ± 100 bp length of sequence from the center of all ChIP-seq peaks and utilized MEME suite²² for motif search. Among 32 antibodies utilized for our ChIP-seq experiments, motif analysis of 22 TFs (Arid3a, Creb3l2, Ctcf, Elf5, Eomes, Ets2, cFos, Hic2, Irf2, Maff, Mafk, Mef2d, Meis1, Pou3f1, Tead4, Tfp2c, Bhlhe40, Cbfa2t3, Foxj2, Id2, and Tbx20) showed either their own canonical motif or motifs highly similar in sequence to the canonical motif as enriched in the

center area of the peaks (**Response Figure 4**). Zfpm1 (Fog1, friend of Gata) is a known interaction partner of Gata3 and its most enriched motif was that of Gata3, suggesting that Zfpm1 binding sites are genuine (**Response Figure 4**). While some TFs (such as Bbx, Fbxo21, Hopx, Lrrfip1, Pcgf5, and Smad6) do not show motif enrichments, the quality of these antibodies were validated by WB or IP-WB (**Response Figure 5**). Four antibodies (H3K4me1, H3K27ac, Med12, and p300) were validated previously (**Response Table 2**). In sum, for almost all TFs we performed ChIP-seq for, we provide evidence satisfying at least one out of three criteria: reference, motif enrichment, and Western blot (**Response Figures 4 and 5, and Response Table 2**)

In multiple places functional and anatomical annotation analysis was performed to infer the function of TFs. However, given that chromatin accessibility tends to reflect cell identity and function, TFs that bind accessible chromatin will always yield results reflecting this regardless of whether those TFs are important in influencing the expression of the presumed target genes. For such annotation enrichment analyses to reflect function I believe that they have to consider the intersect of TF ChIP-seq targets and gene expression changes on TF KD or deletion.

Response (also for the Reviewer's point 2-4 (Q4)): We thank you for this comment. As suggested by the reviewer, we performed RNA-seq upon KD of 7 additional TFs and tested the relationship between target occupancy of total 11 TFs and gene expression changes upon KD of each TF followed by differentiation. We found that the expression patterns of many direct target genes of tested TFs are directly affected by KD upon differentiation observed that the enriched GO terms which are largely consistent with enriched terms of placenta-development-related including placenta development, vascular development, and blood vessel development (**Fig. 5c and Supplementary Figure 7**)

Q6. Are there other experiments that would strengthen the paper further? How much would they improve it, and how difficult are they likely to be?

Questions surrounding antibody specificity need to be addressed, as mentioned above. What experiments are appropriate depend on what data remains after ChIP-seq quality control/validation.

Response: Please see our response to the point 2-5 (Q5).

I would like to see an analysis of the intersect between ChIP-seq data and KD RNA-seq data for the novel TFs in Fig. 5 and some discussion of how this might explain the differentiation phenotypes. I would also ideally like to see a more comprehensive analysis of TF KD across the whole panel of TFs to establish the importance and validity of the conclusions derived from the ChIP-seq data.

Response: We appreciate the reviewer's suggestion and agree that it is an ideal direction to perform a comprehensive analysis of TF KD across the whole panel of TFs. But the current study's purpose is to advance and expedite the research field of placenta biology by providing a useful resource and framework for future studies. To address the reviewer's comment, we have tested 7 additional TFs by KD followed by differentiation

and showed the intersection between TF occupancy and gene expression (please also see our response to the point 2-4 (Q4) and **Fig. 5c and Supplementary Fig. 7**), but we expect that testing whole panel of TFs would be a long-term project, and hope to address this in a separate study.

To establish whether Maff, Mafk, Meis1 and Pou3f1 KD results are likely to reflect *in vivo* reality rather than being an *in vitro* TSC phenomenon I would also like to see *in vivo* expression patterns and a discussion of how the new *in vivo* results relate to the existing *in vitro* data. These experiments/analyses are unlikely to be particularly technically challenging but may be demanding in terms of workforce and expense.

Response (also for the Reviewer's point 3-4): We agree with the reviewer that it is important to examine *in vivo* expression pattern of novel TFs to see whether their expression patterns are correlated with *in vitro* expression data. Therefore, we performed immunohistochemistry (IHC) for multiple TFs in mouse placenta at E15 and added those data to our updated **Fig. 5g**. Consistent with qPCR data, Maff, Mafk, Foxj2, and Ets2 (which are factors whose KD inhibits expression of TGC and SpT marker genes) are highly expressed in the TGC and SpT area of placental sections, while Meis1 and Pou3f1 show weak expression in both TGC and SpT of placenta (**Fig. 5g and Supplementary Fig. 8c**). These data agree with GSEA of RNA-seq data (**Fig. 5e and 5f**) and provide additional evidence that these TFs might be required for the proper development of placenta.

Q7. Are the claims appropriately discussed in the context of previous literature?

Largely, yes. Though it may be appropriate to cite the preexisting literature on TSC enhancer usage and the chromatin landscape as mentioned above. Any new analyses, however, are likely to require reference to more of the previous literature.

Response: We cited the literatures related to enhancer studies in TSCs as the reviewer suggested in the Introduction section (please see our response to the point 2-2 (Q2)).

Q8. If the manuscript is unacceptable in its present form, does the study seem sufficiently promising that the authors should be encouraged to consider a resubmission in the future?

If the authors can address the above then I think a resubmission would be worthwhile, though this seems challenging in a reasonable timeframe.

Response: We again appreciated the reviewer for the constructive suggestions to improve our manuscript. We hope that the revised manuscript will convey our intention more clearly and is more convincing with additional experiments and data analyses.

Further comments:

1. Use of yellow fonts in figures makes the text too hard to read.

Response: We changed the yellow color to orange.

2. The use of the TSC column in the heatmaps in Figure 2 is very helpful but also slightly confusing since it is also red. Would using a different color scheme to the timecourse heatmap be clearer?

Response: As suggested, we used two different color schemes to provide clear comparison.

3. To emphasize or refute the importance of SEs vs REs it would be informative to see the analysis for Extended Data Fig. 2i for REs.

Response (also for the Reviewer points 2-3 (Q3) and 3-1): As suggested, we analyzed the expression of the RE-associated genes in diverse tissues (**Response Figure 3**), confirming that RE-associated genes generally show similar levels of expression across tissues while SE-associated genes show significant differences in their expression across multiple tissues. This means that mapping SEs provides greater power in finding true cell type-specific regulators compared to analyzing REs alone.

4. It would be informative to know what fraction of TF binding events do not coincide with p300. Could the Extended Data Fig. 3f heatmaps be extended to include peaks not at p300 sites?

Response: The main purpose of Extended Data Fig. 3f to show the co-occupancy of p300 with other TFs. We have already presented the information regarding fraction of TF binding events that do not coincide with p300 in the **Supplementary (previously Extended Data) Fig. 3e** of our previous manuscript.

5. How variable is the distribution of TF binding sites for different TFs? Is Extended Data Fig. 3a meaningful?

Response: We have already shown the variability among the binding sites of TFs by showing the correlation of the co-occupancy patterns between TFs in the **Supplementary Fig. 3c**. We think **Supplementary Fig. 3a** is important to show to what extent the regulatory regions distribute across the genome.

6. In the legend for Extended Data Fig. 3d it should be explained how “proximal” and “distal” are defined.

Response: Proximal indicates within ± 2 Kb from the TSS, and distal describes any element further away from the TSS than that. We edited the Figure legend accordingly.

7. There is a misalignment in the Extended Data Fig. 4b axis label.

Response: We corrected the mistake.

8. Tables of read numbers and mapping statistics should be provided to indicate ChIP-seq data quality.

Response: We have already provided requested information in the **Reporting Summary** file.

9. Page 5, paragraph 1 – was 500 μ g of total RNA really used or is it a typographical error? 500ng perhaps? Also referring to volumes of cDNA is not informative without the concentration being clear.

Response: We thank you for pointing this out. It was our mistake and we corrected the unit. We initially used the same amount of total RNA (500 ng) per sample to generate cDNA and assume that synthesized cDNA amount will be similar among samples. To measure the exact amount of cDNA, we have to purify cDNA in order to remove RNAs and proteins, which interfere with measuring the cDNA amount. During this process, we may lose some cDNA. Instead of measuring sample concentrations, we diluted cDNA samples as most of researchers do and use the same volume for each sample to perform qPCR. Levels of Gapdh were used for normalization of the results, and a near-identical Ct for Gapdh was found in qPCR assays, indicating a very similar loading of cDNA.

10. Page 6, para 3 – what version of MACS was used and with what parameters?

Response: We used MACS14 (v1.4.2) with the default peak calling parameters. We updated this information in the **Supplementary Information**.

11. Page 6, para 4 – can a rationale for the window parameters e.g. 20kb upstream be provided?

Response: Please see our response to the point 1-2.

12. Page 6, para 5 – I think it should read “...website of the Young lab...”.

Response: Corrected.

13. Page 7, para 1 – can taking the top and bottom 10% of genes from each dataset rather than making a judgment based on statistical significance be justified?

Response: When we randomly select one gene, the probability that the selected gene belongs to the top 10% is 0.1. The probability that a gene chosen randomly belongs to top 10% at least 3 times among 5 expression data sets is sum of probability (0.00081 (3 chosen) + 0.00009 (4 chosen) + 0.00001 (5 chosen) = 0.00091). We found

77 up-regulated and 45 down-regulated genes using top 10% or bottom 10% criteria explained above. The hypergeometric P -value of 77 up-regulated and 45 down-regulated genes are $P < 0.001$, which is statistically significant.

14. Page 7, para 3 – what versions and parameters were used for GREAT and DAVID

Response: We used GREAT (v3.0.0) and DAVID (v6.7). We added all these information in the Methods section.

Reviewer #3:

Understanding the regulatory networks that control trophoblast development and its differentiation to the different lineages that conform the mature placenta is of great interest but, as the authors point out, has been less extensively studied than other cell populations of the early mammalian embryos that contribute to the embryo proper. In this regard, the work presented here by Lee and colleagues is of clear interest and could help to complement our knowledge of this process.

In order to do so, the authors carry out an extensive genomic characterization of trophoblast stem cells (TSCs) used as a proxy for trophoblast lineage development. Using a combination of ChIP-seq, RNA-seq and ATAC-seq on multipotent and differentiated TSCs, together with knock-down of candidate genes, the authors describe some novel transcriptional regulators that could be involved in TSC maintenance and differentiation.

Overall, this data-rich manuscript provides a very useful resource for the community that will sure benefit from the different genomic analysis performed on TSCs. However, despite the impressive amount of data and analysis, this work falls short of providing a convincing physiological context for the novel findings obtained in a culture system. Major issues in this regard are discussed below.

Response: We would like to thank the reviewer for constructive comments and helpful suggestions. We contend that our manuscript has benefited greatly from implementing reviewer's suggested changes and hope that the reviewer finds our responses clear and concise.

1. The authors use the characterization of super-enhancers in TSCs to identify novel gene involved in TE and placenta biology. Although the description of super-enhancers in another cell type is interesting, in this work there is little novel insight into the nature and function of these regulatory elements. For sure this is not the aim of the work, but the value of using super-enhancers to identify novel genes important in TSCs is not properly justified. Couldn't the authors have used regular enhancers equally? Or simply differential gene expression? The fact that the authors link more than one thousand genes to TSC super-enhancers would argue that this approach is not that useful to pinpoint key regulators.

Response (also for the Reviewer points 1-1 and 2-3 (Q3)): As the reviewer mentioned, it is also feasible to identify cell type-specific TFs based on REs or gene expression profiling. However, we believe that SE-guided prediction of cell-type specific TFs is superior to other approaches, especially when combined with other approaches such as gene expression profiling. Most of studies revealed that tens of thousands of REs are utilized in any given cell type; thus, there may be a high false positive rate when attempting to predict key regulators by mapping REs alone. As we showed in the **Fig. 2**, differential gene expression-based approach cannot easily predict multiple classes of putative regulators. Moreover, many known key TFs or important regulators are SE-associated factors in various other cellular contexts^{1,11,12} from which we developed our approach. While we sincerely agree with the importance of RE-associated genes in cell type-specific gene regulation, our intention was to focus on the utility of multiple enhancer-associated loci for prediction of key regulators. We provide boxplot data from the analysis of RE-associated genes in various tissues (**Response Figure 3**), showing that expression patterns of RE-associated genes are not significantly different among multiple tissues while SE-associated genes show significant differences among different tissue types. Nonetheless, to address reviewers' concern, we toned down our emphasis on SEs in multiple places and described importance in RE-related factors in modulating

cellular identity. We also provide the list of RE-associated genes in **Supplementary Table 4** in our revised manuscript, hoping that the revised manuscript convey our intention clearly.

Despite the strong emphasis on super-enhancers, we do not learn much about their function or role in gene regulation more than correlative analysis. The authors claim cooperativity and synergistic effects of transcriptional inputs on these regulatory elements, but this is based only from the presence of putative TF binding sites on the super-enhancers and in some cases co-occupancy as determined by ChIP-seq. Taking into account that the size of super-enhancers ranges from 1 up to 135 kb, presence of a consensus binding site sequence cannot be taken as evidence of regulation by a given factor. No functional assays are provided to prove that these elements are having a role in TSCs. Without this sort of evidence, all analysis shown are merely correlative and conclusions based on super-enhancer predictions should be toned down.

Response: We apologize for causing confusion. For the motif analyses we did not use entire SE area. Since most SEs comprises multiple enhancers, we extracted ± 100 bp sequence of the center of each peak within the SEs and performed motif analysis. Therefore, the motifs are enriched at the binding sites of each TF tested. As commented, we did not provide direct evidence of SE function aside from gene expression correlation and motif enrichment. We therefore toned down the importance of SEs throughout the manuscript. In addition, we tested the average target gene expression and the level of co-occupancy of the TFs we tested (**Response Figure 6**), suggesting that multiple TFs cooperatively and synergistically affect the expression of target genes.

2. The authors compare enhancer predictions and several TF occupancy patterns between multipotent and differentiated TSCs. Here, little detail is provided regarding the differentiation protocol and the resulting cell types. Differentiation was induced by removing Fgf4 and heparin, what has been described to results mainly in differentiation of trophoblast giant cells but not other trophoblast-derived cell types (reviewed in Latos & Hemberger, 2016, Development 143, 3650-60). Therefore, without further characterization, we do not really know what cell types are obtained. No time course or marker expression is shown in order to more carefully understand the system used to analyze differentiating TSCs.

Response: We appreciate the reviewer for raising this important point, which we did not clearly address in the original manuscript. At this point, a method promoting differentiation of mouse TSCs into a specific trophoblast cell type has not been well established²³. While some previous literature like the reviewer mentioned describe limited differentiation of TSCs toward TGCs upon withdrawal of Fgf4 and heparin, other prior studies of marker gene expression showed that subpopulation of TSCs can be also differentiated into SpT and labyrinthine trophoblast cells fates by withdrawal of Fgf4, heparin, and conditioned media²⁴ which is a commonly used differentiation method²⁵. We also confirmed that the study of time-course differentiation of TSCs into subtype cells consisting placenta²⁶ showed that SpT and SynT lineage markers are induced within 3 days of differentiation (**Response Figure 7**). Consistent with these prior works, we could detect significant induction of spongiotrophoblast (SpT) marker genes (Ascl2, Tssc3, Cea4, Tpbpa) and syncytiotrophoblast (SynT) marker genes (SynA, SynB, and Gcm1) upon differentiation of TSCs as shown in **Fig. 5a**.

3. This problem is most clearly shown in Fig. 5a, where apparently after 3 days, markers for all different cell types examined are present simultaneously. Furthermore, using a single marker for a specific subtype, as is the case of *Gcm1* for SynT, is rather weak to drive any definitive conclusions. This panel is used then to describe the effect of the knock-down of a number of newly identified transcription factors that could be involved in TSC differentiation (Fig 5b). According to the description of this experiment in Methods, RT-qPCR was performed at least in duplicates, and changes are shown as fold changes. Although the results seem robust, no statistical analysis is provided for this experiment. RNA-seq for a number of these KD was generated, so more quantitative data should be available. Are the changes depicted in Fig. 5b statistically significant in standard differential gene expression analysis of RNA-seq data for *Maff*, *Mafk*, *Meis1* or *Pou3f1* KDs?

Response (also for the Reviewer's point 1-3 in part): We agree with reviewer's comment that relying on one marker gene expression is weak evidence to determine the specific lineage, therefore we performed qPCR for additional SynT markers, such as *SynA* and *SynB*. *Gcm1* and *SynB* are markers of syncytiotrophoblast layer II (SynT-II) and *SynA* is a marker of layer I (SynT-I)²⁷. We found that *SynB* and *Gcm1* show a similar expression pattern when TSC-specific TFs are depleted, while *SynA* show an opposite expression pattern. This suggest that the multiple TFs (*Elf5*, *Hopx*, *Id2*, *Pou3f1*, and *Zfpm1*) promotes SynT II differentiation while inhibits SynT-I differentiation. As we observe the differential expression of markers for SynT, we updated our figure shown in Fig. 5b, accordingly. As the reviewer's suggestion, we perform additional RNA-seq experiments and analyzed the data, confirming that the majority of differentially expressed genes in qPCR are statistically significant in RNA-seq data (Response Figure 8).

4. The most interesting and central claim of this manuscript is the identification of novel transcription factors with a previously undescribed role in TSCs, what would extend our knowledge of the gene regulatory networks underlying TE and placental development. Starting from more than 150 transcription factors initially identified in their super-enhancer based screen, the authors examine genome-wide binding of 27 of these in TSCs and of a subset in differentiated TSCs. The logic for the selection of these factors is unclear and should be described (antibody availability for ChIP-seq?). Network analysis of this data leads the authors to further select a handful of these for KD experiments, leading to define novel regulatory interactions as depicted in Fig. 5e.

Response: We are sorry for the lack of clarity behind why we selected those subsets of candidates for ChIP-seq. We were particularly interested in TSC-specific TFs that show expression changes during TSC differentiation. Thus, we chose TFs mainly from the Classes 1 to 3, and we selected TFs that we had high-quality antibodies for. For the KD approaches, we randomly selected multiple known and unknown TFs from the Classes 1 to 3.

Data shown to support these claims seems at most preliminary, and further insight should be provided. For example, the effect of combinatorial KD of these factors would shed light on their supposed cooperativity and synergistic action. Most importantly, the physiological and *in vivo* relevance of the findings needs to be proven. At least, the description of the expression patterns of a number of these genes during placenta development should be provided to support them having a role in this process. Re-examining mutant phenotypes would also be desirable, although understandably this would be out of the scope of the present manuscript.

Response (also for the Reviewer's point 2-6 (Q6): We agree with the reviewer that it is important to examine physiological and *in vivo* relevance of the novel TFs. To test whether the expression patterns of candidate TFs in the placenta are correlated with *in vitro* data, we performed immunohistochemistry (IHC) for multiple TFs in mouse placenta and updated those data in Fig. 5g. Consistent with qPCR data, *Maff*, *Mafk*, *Foxj2*, and *Ets2*,

whose KD inhibits expression of TGC and SpT marker genes, are highly expressed in the TGC and SpT area of placental sections (obtained from E15) while Meis1 and Pou3f1 show weak and partial expression in both TGC and SpT regions of placenta. These data agree with GSEA of RNA-seq data (**Fig. 5e and 5f**) and provide additional evidence that these TFs are associated with specific placental lineages. While additional functional validation *in vivo* or re-examining of KO phenotypes is desirable, as the reviewer commented, we expect that *in vivo* studies sufficient to demonstrate the developmental roles of key TFs would require a considerably greater investment of time and effort, and we hope to address this issue in a future study.

Response Table 1. Percentage of a TF's binding sites that are assigned to genes by proximal-distal looping interactions

TFs	% distal interaction	% no interaction
Arid3a	52.1	47.9
Bbx	23.8	76.2
Bhlhe40	30.0	70.0
Ctcf	61.2	38.8
Cbfa2t3	38.7	61.3
Creb3l2	34.2	65.8
Elf5	51.2	48.8
Eomes	41.6	58.4
Ets2	40.3	59.7
Fbxo21	45.7	54.3
Foxj2	53.6	46.4
Hic2	54.3	45.7
Hopx	15.4	84.6
Id2	47.3	52.7
Irf2	29.2	70.8
Lrrfip1	16.0	84.0
Maff	52.1	47.9
Mafk	46.9	53.1
Med12	49.5	50.5
Mef2d	49.4	50.6
Meis1	57.4	42.6
Pcgf5	39.5	60.5
Pou3f1	50.3	49.7
Smad6	50.9	49.1
Tbx20	14.7	85.3
Tead4	55.9	44.1
Tfap2c	51.2	48.8
Zfp1	53.6	46.4
cFos	59.7	40.3
p300	54.3	45.7
Average	44.0	56.0

Response Table 2. Validation of antibodies

NO	Gene	Comp	Source	M.W	Cat #	Validation or ChIP-grade evidence	Motif	Western
1	Arid3a	SC	goat	64	NB100-279	https://www.encodeproject.org/antibodies/ENCAB000ASP/	Yes	
2	Bbx	SC	rabbit	105	sc-99279X		not enriched	Yes
3	Bhlhe40	SC	mouse	45	sc-101023	https://www.encodeproject.org/antibodies/ENCAB000AEJ/	Yes (Similar motif)	
4	Cbfa2t3	Abcam	rabbit	71	ab33072		Yes (Similar motif)	Yes
5	Creb3l2	SC	rabbit	57	sc-366044		Yes	Yes
6	Ctcf	Millipore	rabbit		07-729	https://www.encodeproject.org/antibodies/ENCAB830JLB/	Yes	
7	Elf5	SC	goat	30	sc-9645X	PMID:23300383	Yes	
8	Eomes	SC	rabbit	73	sc-98555 X		Yes	
9	Ets2	SC	rabbit	55	sc-351X	PMID:25993293	Yes	
10	Fbxo21	Abcam	rabbit	71	ab179818		not defined	Yes
11	cFos	SC	rabbit	41	sc-7202X	https://www.encodeproject.org/antibodies/ENCAB000AEQ/	Yes	
12	Foxj2	SC	goat	63	sc-54374		Yes (Similar motif)	Yes
13	H3K27ac	abcam	rabbit		AB4729	https://www.encodeproject.org/antibodies/ENCAB000APD/	not defined	
14	H3K4me1	abcam	rabbit		ab8895	https://www.encodeproject.org/antibodies/ENCAB969VGQ/	not defined	
15	Hic2	SC	goat	66	sc-86486 X		Yes	
16	Hopx	SC	rabbit	8	sc-30216		not defined	Yes
17	Id2	SC	rabbit	15	sc-489 X		Yes (Similar motif)	
18	Irf2	SC	rabbit	39	sc-498X		Yes	
19	Lrrfip1	SC	rabbit	79	sc-68387X		not defined	Yes
20	Maff	SC	rabbit	17	sc-22831X		Yes	
21	Mafk	SC	rabbit	18	sc-477X	https://www.encodeproject.org/antibodies/ENCAB000AIK/	Yes	
22	Med12	Bethyl	rabbit	244	A300-774A	PMID:25083870	not defined	
23	Mef2d	SC	mouse	55	sc-271153X		Yes	Yes
24	Meis1	SC	goat	43	sc-10599X	PMID:24760698	Yes	
25	p300	SC	rabbit	300	sc-585	https://www.encodeproject.org/antibodies/ENCAB000AJM/	not defined	
26	Pcgf5	Abcam	mouse	30	ab76724		not defined	Yes
27	Pou3f1	SC	goat	45	sc-11661X		Yes	Yes
28	Smad6	SC	goat	54	sc-7004X		not defined	Yes
29	Tbx20	SC	goat	33	sc-18486X		Yes (Similar motif)	
31	Tead4	abcam	mouse	48	ab58310	PMID:26923725	Yes	
30	Tfap2c	SC	rabbit	49	sc-8977	https://www.encodeproject.org/antibodies/ENCAB000AMZ/	Yes	
32	Zfp1	SC	goat	105	sc-9361X		Gata3	

REFERENCES

- 1 Hnisz, D. *et al.* Super-enhancers in the control of cell identity and disease. *Cell* **155**, 934-947, doi:10.1016/j.cell.2013.09.053 (2013).
- 2 Natarajan, A., Yardimci, G. G., Sheffield, N. C., Crawford, G. E. & Ohler, U. Predicting cell-type-specific gene expression from regions of open chromatin. *Genome Res* **22**, 1711-1722, doi:10.1101/gr.135129.111 (2012).
- 3 Fujiwara, T. *et al.* Discovering hematopoietic mechanisms through genome-wide analysis of GATA factor chromatin occupancy. *Mol Cell* **36**, 667-681, doi:10.1016/j.molcel.2009.11.001 (2009).
- 4 Yu, M. *et al.* Insights into GATA-1-mediated gene activation versus repression via genome-wide chromatin occupancy analysis. *Mol Cell* **36**, 682-695, doi:10.1016/j.molcel.2009.11.002 (2009).
- 5 Lieberman-Aiden, E. *et al.* Comprehensive mapping of long-range interactions reveals folding principles of the human genome. *Science* **326**, 289-293, doi:10.1126/science.1181369 (2009).
- 6 Dixon, J. R. *et al.* Topological domains in mammalian genomes identified by analysis of chromatin interactions. *Nature* **485**, 376-380, doi:10.1038/nature11082 (2012).
- 7 Schoenfelder, S. *et al.* Divergent wiring of repressive and active chromatin interactions between mouse embryonic and trophoblast lineages. *Nat Commun* **9**, 4189, doi:10.1038/s41467-018-06666-4 (2018).
- 8 Nishioka, N. *et al.* Tead4 is required for specification of trophoblast in pre-implantation mouse embryos. *Mech Dev* **125**, 270-283, doi:10.1016/j.mod.2007.11.002 (2008).
- 9 Buenrostro, J. D., Wu, B., Chang, H. Y. & Greenleaf, W. J. ATAC-seq: A Method for Assaying Chromatin Accessibility Genome-Wide. *Curr Protoc Mol Biol* **109**, 21-29, doi:10.1002/0471142727.mb2129s109 (2015).
- 10 McLean, C. Y. *et al.* GREAT improves functional interpretation of cis-regulatory regions. *Nat Biotechnol* **28**, 495-501, doi:10.1038/nbt.1630 (2010).
- 11 Whyte, W. A. *et al.* Master transcription factors and mediator establish super-enhancers at key cell identity genes. *Cell* **153**, 307-319, doi:10.1016/j.cell.2013.03.035 (2013).
- 12 Vahedi, G. *et al.* Super-enhancers delineate disease-associated regulatory nodes in T cells. *Nature* **520**, 558-562, doi:10.1038/nature14154 (2015).
- 13 Boyer, L. A. *et al.* Core transcriptional regulatory circuitry in human embryonic stem cells. *Cell* **122**, 947-956, doi:10.1016/j.cell.2005.08.020 (2005).
- 14 Loh, Y. H. *et al.* The Oct4 and Nanog transcription network regulates pluripotency in mouse embryonic stem cells. *Nat Genet* **38**, 431-440, doi:ng1760 [pii]
- 15 Kim, J., Chu, J., Shen, X., Wang, J. & Orkin, S. H. An extended transcriptional network for pluripotency of embryonic stem cells. *Cell* **132**, 1049-1061, doi:10.1016/j.cell.2008.02.039 (2008).
- 16 Chen, X. *et al.* Integration of external signaling pathways with the core transcriptional network in embryonic stem cells. *Cell* **133**, 1106-1117, doi:S0092-8674(08)00617-X [pii]
- 17 Scacheri, P. C. *et al.* Genome-wide analysis of menin binding provides insights into MEN1 tumorigenesis. *PLoS Genet* **2**, e51, doi:10.1371/journal.pgen.0020051 (2006).
- 18 Yang, A. *et al.* Relationships between p63 binding, DNA sequence, transcription activity, and biological function in human cells. *Mol Cell* **24**, 593-602, doi:10.1016/j.molcel.2006.10.018 (2006).
- 19 Krig, S. R. *et al.* Identification of genes directly regulated by the oncogene ZNF217 using chromatin immunoprecipitation (ChIP)-chip assays. *J Biol Chem* **282**, 9703-9712, doi:10.1074/jbc.M611752200 (2007).
- 20 Martone, R. *et al.* Distribution of NF-kappaB-binding sites across human chromosome 22. *Proc Natl Acad Sci U S A* **100**, 12247-12252, doi:10.1073/pnas.2135255100 (2003).
- 21 Farnham, P. J. Insights from genomic profiling of transcription factors. *Nat Rev Genet* **10**, 605-616, doi:10.1038/nrg2636 (2009).
- 22 Bailey, T. L., Johnson, J., Grant, C. E. & Noble, W. S. The MEME Suite. *Nucleic Acids Res* **43**, W39-49, doi:10.1093/nar/gkv416 (2015).
- 23 Latos, P. A. & Hemberger, M. From the stem of the placental tree: trophoblast stem cells and their progeny. *Development* **143**, 3650-3660, doi:10.1242/dev.133462 (2016).

- 24 Hughes, M. *et al.* The Hand1, Stra13 and Gcm1 transcription factors override FGF signaling to promote terminal differentiation of trophoblast stem cells. *Dev Biol* **271**, 26-37, doi:10.1016/j.ydbio.2004.03.029 (2004).
- 25 Kidder, B. L. Derivation and manipulation of trophoblast stem cells from mouse blastocysts. *Methods Mol Biol* **1150**, 201-212, doi:10.1007/978-1-4939-0512-6_13 (2014).
- 26 Ralston, A. *et al.* Gata3 regulates trophoblast development downstream of Tead4 and in parallel to Cdx2. *Development* **137**, 395-403, doi:137/3/395 [pii]
- 27 Simmons, D. G. *et al.* Early patterning of the chorion leads to the trilaminar trophoblast cell structure in the placental labyrinth. *Development* **135**, 2083-2091, doi:10.1242/dev.020099 (2008).

Reviewers' comments:

Reviewer #1 (Remarks to the Author):

The authors have addressed most of the key points I raised, and for the most part I am convinced that the manuscript's finding and claims are robust.

There is, however, one important point that was not properly addressed. On the issue of sequencing data alignment, the authors confirmed that the default settings of bowtie2 were used. We have extensive experience in this and can assure the authors that these settings do not yield uniquely mapped reads. As per the authors' reply based on the bowtie2 manual, reads that align equally well to more than one location (non-unique reads) are assigned to one of those locations at random. The data therefore include non-unique reads with randomly assigned locations, which may yield false peaks. To obtain uniquely mapped reads the data need to be filtered. Whilst I appreciate that this is a cumbersome task given the amount of data produced for this study, all alignments and peak calling should be performed on uniquely mapped data.

The new IHC data are interesting. However, the description of the results give the impression that MAFF, MAFK, ETS2 and FOXJ2 expression is in some way specific to the TGC and spongiotrophoblast layers, whereas they are clearly also expressed in the labyrinth. This should be made clear. It would also be reassuring if there were areas (perhaps in the embryo) where there was no staining, to ensure that these antibodies are not just 'sticky' in an IHC context.

Minor: supplementary table 7 is very useful, but has not been mentioned in the main text.

Miguel Branco

Reviewer #2 (Remarks to the Author):

The authors have made a strong effort to address reviewers' comments including valuable additional data and analyses. Regarding the antibody validation - I am not suggesting any further work in this

area. However, ideal validation involves knockdown/mutant samples to demonstrate specificity, and IP-Westerns to show that antibodies IP as well as recognise denatured protein. Given the scale of the study, in my opinion the validation in the rebuttal is appropriate, and sufficient for readers to form a clear view of the likely reliability to the data. In the event of publication I think it is very important that the rebuttal and associated figures are also published, and that the main manuscript would benefit from signposting to the "Peer Review File" in the Methods section if permitted.

Minor comment:

Pages 8 & 9 of the supp PDF file - should references to a "RefFat file" actually be RefFlat?

Reviewer #3 (Remarks to the Author):

In this revised version, the authors have addressed some of the concerns raised by this and the other reviewers, but however other have not been properly addressed.

In the rebuttal, the authors argue extensively for their use of super-enhancers in the analysis instead or regular enhancers. Certainly, using this as a way to select a subgroup of enhancers seems valid. In this regard, the evidence provided in Response Figure 3 would be the most telling, and as such would merit being included in full in the supplementary information accompanying this paper.

There is still a lack of information regarding the differentiation of TSCs. For example, Response Figure 7 does not show a marker for TGCs, which presumably will be the most abundant cell type present in the culture. Furthermore, the differences in levels of expression of SpT and SynT markers shown in this graph do not appear to be that impressive.

The authors show in Response Figure 8 z-scores, but do these represent statistically significant changes in standard RNA-seq conditions (adj pval<0,05)? As far as I understand, the data shown in Fig. 5b of the revised version are still based on at least 2 duplicates, with no statistical analysis.

Cooperativity of different transcription factors in regulating target genes is still far from being proven in the revised version. The evidence presented in Response Figure 6 does not prove so. Was this data corrected for super-enhancer length for each of the genes? Furthermore, the authors do not provide any answer to the suggestion of performing combined knock-downs of different factors to prove cooperative or synergistic activity.

The quality of the immunos shown in Figure 5 is extremely poor. Compared to those of Proliferin and Tpbpa shown in Supplementary Figure 8, they look like background. No differences can be observed between the different layers. Was any check done for the specificity of the antibodies? Evidence that they are working properly needs to be provided, for example by including a control tissue where expression of the markers is known to be present. If not possible, in situ hybridization on sections should be performed. As such, this data is not valid.

Reviewer #1 (Remarks to the Author):

The authors have addressed most of the key points I raised, and for the most part I am convinced that the manuscript's finding and claims are robust.

1. There is, however, one important point that was not properly addressed. On the issue of sequencing data alignment, the authors confirmed that the default settings of bowtie2 were used. We have extensive experience in this and can assure the authors that these settings do not yield uniquely mapped reads. As per the authors' reply based on the bowtie2 manual, reads that align equally well to more than one location (non-unique reads) are assigned to one of those locations at random. The data therefore include non-unique reads with randomly assigned locations, which may yield false peaks. To obtain uniquely mapped reads the data need to be filtered. Whilst I appreciate that this is a cumbersome task given the amount of data produced for this study, all alignments and peak calling should be performed on uniquely mapped data.

Response: We are grateful for the reviewer's comments and overall support of our manuscript. As requested, we conducted peak calling after filtering out non-uniquely mapped reads, and found that the global correlation between the previously processed vs. newly processed data was largely similar (**Revision 2 Response Figure 1**). Although this filtering does not significantly affect the main claim of the manuscript, we regenerated and updated all associated figures (marked in red in each panel) and supplemental data using the newly processed data. We also uploaded newly processed data onto GEO under the accession number GSE110950 (secure token for review: krevgkktjqpzpkp)

2. The new IHC data are interesting. However, the description of the results give the impression that MAFF, MAFK, ETS2 and FOXJ2 expression is in some way specific to the TGC and spongiotrophoblast layers, whereas they are clearly also expressed in the labyrinth. This should be made clear. It would also be reassuring if there were areas (perhaps in the embryo) where there was no staining, to ensure that these antibodies are not just 'sticky' in an IHC context.

Response (see also the response to the reviewer 3's comment 5): Indeed, we observed detectable expression of the 4 factors (Maff, Mafk, Ets2, and Foxj2) within the labyrinthine layer of placenta. However, the signal in SynT was weaker relative to the signal detected in TGC or SpT. Compared to TGC and SpT markers, changes in the levels of SynT-specific markers were not significant or consistent upon KD of these factors followed by differentiation (**Figure 5b**). Therefore, we did not strongly claim an important role for these factors in SynT or labyrinthine layer development in the text. Following the suggestion of the reviewer, we performed IHC with mouse embryos in order to assure the quality of antibodies used. As shown in **Revision 2 Response Figure 2** (newly added **Supplementary Fig. 9b**), we confirmed that our antibodies bound to known specific tissue loci of the mouse embryo, suggesting that they do not bind everywhere non-specifically. Moreover, enriched motifs of these TFs from ChIP-seq data as well as Western blot of Foxj2 supported the specificity of these antibodies (newly added **Supplementary Fig. 10a and 10b** in response to the reviewer 2's comment 1).

Minor: supplementary table 7 is very useful, but has not been mentioned in the main text.

Response: As commented, we added a sentence referencing **Supplementary Table 7** in page 6 in the main text. Now it reads as “Individual TFs occupy as many as tens of thousands of targets including the genes regulated by promoter-enhancer looping (**Supplementary Fig. 3b, Supplementary Table 7**).”

Reviewer #2 (Remarks to the Author):

1. The authors have made a strong effort to address reviewers' comments including valuable additional data and analyses. Regarding the antibody validation - I am not suggesting any further work in this area. However, ideal validation involves knockdown/mutant samples to demonstrate specificity, and IP-Westerns to show that antibodies IP as well as recognize denatured protein. Given the scale of the study, in my opinion the validation in the rebuttal is appropriate, and sufficient for readers to form a clear view of the likely reliability to the data. In the event of publication I think it is very important that the rebuttal and associated figures are also published, and that the main manuscript would benefit from signposting to the "Peer Review File" in the Methods section if permitted.

Response: We are grateful for the reviewer's overall support of our manuscript. We have updated our antibody validation procedures and results in Supplementary Methods (page 7) and newly added Supplementary Fig. 10 as well as Supplementary Table 10.

Minor comment:

Pages 8 & 9 of the supp PDF file - should references to a "RefFat file" actually be RefFlat?

Response: We have corrected these errors.

Reviewer #3 (Remarks to the Author):

1. In this revised version, the authors have addressed some of the concerns raised by this and the other reviewers, but however other have not been properly addressed.

In the rebuttal, the authors argue extensively for their use of super-enhancers in the analysis instead of regular enhancers. Certainly, using this as a way to select a subgroup of enhancers seems valid. In this regard, the evidence provided in Response Figure 3 would be the most telling, and as such would merit being included in full in the supplementary information accompanying this paper.

Response: As requested, we included our previous **Response Figure 3** in the current **Supplementary Fig. 2I**

2. There is still a lack of information regarding the differentiation of TSCs. For example, Response Figure 7 does not show a marker for TGCs, which presumably will be the most abundant cell type present in the culture. Furthermore, the differences in levels of expression of SpT and SynT markers shown in this graph do not appear to be that impressive.

Response: As we mentioned in our previous response, we agree with the reviewer that TGC is the major cell type during differentiation of mTSCs. The induction of TGC marker genes was obvious upon differentiation of TSCs, therefore, we did not include TGC marker genes in our previous **Response Figure 7**. To accommodate the reviewer's concern about the degree of SpT and SynT marker expression, we have downloaded another time-course TSC differentiation data (GSE18507) (Kidder and Palmer, 2010) and added TGC marker gene expression in **Revision 2 Response Figure 3**. In regard to the reviewer's first concern, the differentiation method we used for TSC is a published and standard method of TSC differentiation that has been utilized by several labs to examine the changes of different lineage marker genes, including SpT and SynT (Hayes and Naegeli, 2010; Hughes et al., 2004; Kidder, 2014; Kidder and Palmer, 2010; Ralston et al., 2010; Wahl, 2018). In **Fig. 5a**, we clearly showed a significant induction of SynT and SpT maker genes upon differentiation of TSCs.

3. The authors show in Response Figure 8 z-scores, but do these represent statistically significant changes in standard RNA-seq conditions (adj pval<0,05)? As far as I understand, the data shown in Fig. 5b of the revised version are still based on at least 2 duplicates, with no statistical analysis.

Response: We apologize for the lack of clarity. Z-score of 1.96 represents *P*-value of 0.05. The vast majority of genes showed more than Z-score of 2, indicating the significant changes of marker gene expression upon KD of TSC-specific TFs. To further clarify our claim, we performed an additional replicate experiment for **Fig. 5b** (now n=3) and tested statistical significance using Student's t-test. As shown in **Revision 2 Response Figure 4**, most of the tested genes are statistically significant.

4. Cooperativity of different transcription factors in regulating targets genes is still far from been proven in the revised version. The evidence presented in Response Figure 6 does not prove so. Was this data corrected for super-enhancer length for each of the genes? Furthermore, the authors do not provide any answer to the suggestion of performing combined knock-downs of different factors to prove cooperative or synergistic activity.

Response: We agree to the reviewer’s point and our previous **Response Figure 6** did not directly indicate synergistic effects of TFs. However, given the fact that TF binding to the regulatory region of a gene implies regulatory control, a positive correlation between the number of TFs associated and target gene expression implies a cooperative action of multiple TFs (Hannenhalli and Levy, 2002). Regarding the previous **Response Figure 6**, we apologize for the lack of clarity. The data represent the general relationship between the numbers of TFs that bind to their common target genes and average expression levels of the genes. For each gene, we searched ± 20 Kb from the TSS and gene body area, so super-enhancer length was not a necessary consideration for this analysis. Following the reviewer’s suggestion to test the cooperative or synergistic effects of TFs, we performed double KD of Pou3f1 and Meis1 (Class 2 TFs) as well as Maff and Mafk (Class 3 TFs) followed by differentiation. As shown in **Revision 2 Response Figure 5**, compared to single KD, double KD in general shows a stronger influence on marker gene expression. However, double KD effects are somewhat various on different target genes and overall the effects of double KD seem to be additive rather than synergistic. We therefore removed the word “synergistic” in the text at page 8. Now it reads as “Fifth, the TFs in the network function together to promote gene expression”.

5. The quality of the immunos shown in Figure 5 is extremely poor. Compared to those of Proliferin and Tpbpa shown in Supplementary Figure 8, they look like background. No differences can be observed between the different layers. Was any check done for the specificity of the antibodies? Evidence that they are working properly needs be provided, for example by including a control tissue where expression of the markers is known to be present. If not possible, in situ hybridization on sections should be performed. As such, this data is not valid.

Response (see also our response to the reviewer 1’s comment 2): As suggested, we performed IHC in mouse embryo sections to validate the antibody specificity. As shown in **Revision 2 Response Figure 2** (newly added **Supplementary Fig. 9b**), each antibody recognizes a specific tissue of mouse embryo section. For example, Ets2 and Foxj2 antibodies can recognize pectoralis (Ristevski et al., 2002) and carotid artery (Visel et al., 2004), respectively. Our Maff antibody can specifically recognize dorsal root ganglia, while the Mafk antibody can detect the neopallial cortex (Onodera et al., 1999). Cumulatively, this IHC data supports the specificity of the antibodies that we used.

Revision 2 References

- Hannenhalli, S., and Levy, S. (2002). Predicting transcription factor synergism. *Nucleic acids research* 30, 4278-4284.
- Hayes, R.P., and Naegeli, A.N. (2010). The contribution of pretreatment expectations and expectation-perception difference to change in treatment satisfaction and end point treatment satisfaction in the context of initiation of inhaled insulin therapy in patients with type 2 diabetes. *Diabetes technology & therapeutics* 12, 447-453.

Hughes, M., Dobric, N., Scott, I.C., Su, L., Starovic, M., St-Pierre, B., Egan, S.E., Kingdom, J.C., and Cross, J.C. (2004). The Hand1, Stra13 and Gcm1 transcription factors override FGF signaling to promote terminal differentiation of trophoblast stem cells. *Developmental biology* 271, 26-37.

Kidder, B.L. (2014). Derivation and manipulation of trophoblast stem cells from mouse blastocysts. *Methods in molecular biology* 1150, 201-212.

Kidder, B.L., and Palmer, S. (2010). Examination of transcriptional networks reveals an important role for TCFAP2C, SMARCA4, and EOMES in trophoblast stem cell maintenance. *Genome research* 20, 458-472.

Onodera, K., Shavit, J.A., Motohashi, H., Katsuoka, F., Akasaka, J.E., Engel, J.D., and Yamamoto, M. (1999). Characterization of the murine mafF gene. *The Journal of biological chemistry* 274, 21162-21169.

Ralston, A., Cox, B.J., Nishioka, N., Sasaki, H., Chea, E., Rugg-Gunn, P., Guo, G., Robson, P., Draper, J.S., and Rossant, J. (2010). Gata3 regulates trophoblast development downstream of Tead4 and in parallel to Cdx2. *Development* 137, 395-403.

Ristevski, S., Tam, P.P., Hertzog, P.J., and Kola, I. (2002). Ets2 is expressed during morphogenesis of the somite and limb in the mouse embryo. *Mechanisms of development* 116, 165-168.

Visel, A., Thaller, C., and Eichele, G. (2004). GenePaint.org: an atlas of gene expression patterns in the mouse embryo. *Nucleic acids research* 32, D552-556.

Wahl, D. (2018). "Antiphospholipids": the more, the worse? *Blood* 131, 2092-2094.

REVIEWERS' COMMENTS:

Reviewer #1 (Remarks to the Author):

The authors have addressed my queries. I particularly commend the huge effort to re-do all figures with uniquely mapped data.

Miguel Branco

Reviewer #3 (Remarks to the Author):

The authors have made a very important effort to address all issues raised by the reviewers, by re-analysis of the data, performing new experiments or increasing the number of replicates where needed. Therefore, this data-rich manuscript provides now robust findings and the main claims are very well supported by the results.

Regarding the immunos shown in Fig. 5, the new evidence provided to show the specificity of the antibodies used addresses the concerns previously raised by this reviewer. However, the authors should more clearly state in the Results section that despite the specificity shown by the antibodies in other sites of known expression, staining in placenta does not show a clear layer-specific pattern. The authors should also describe in more detail the results of the control staining shown in Supplementary Fig 9b, and include in the main text or in the legend the references for these specific patterns of expression, that they do include in the rebuttal.

REVIEWERS' COMMENTS:

Reviewer #1 (Remarks to the Author): The authors have addressed my queries. I particularly commend the huge effort to re-do all figures with uniquely mapped data.

Response: We appreciated the reviewer for the constructive comments.

Reviewer #3 (Remarks to the Author): The authors have made a very important effort to address all issues raised by the reviewers, by re-analysis of the data, performing new experiments or increasing the number of replicates where needed. Therefore, this data-rich manuscript provides now robust findings and the main claims are very well supported by the results.

Regarding the immunos shown in Fig. 5, the new evidence provided to show the specificity of the antibodies used addresses the concerns previously raised by this reviewer. However, the authors should more clearly state in the Results section that despite the specificity shown by the antibodies in other sites of known expression, staining in placenta does not show a clear layer-specific pattern.

Response: We have added a new sentence to address the reviewer's concern at the page 11 in the main text: "Unexpectedly, we also observed some weak expression signals of those 4 genes in labyrinthine area despite of the confirmed specificity of antibodies we used (Supplementary Fig. 9b), implying that they might be also implicated in development of labyrinthine."

The authors should also describe in more detail the results of the control staining shown in Supplementary Fig 9b, and include in the main text or in the legend the references for these specific patterns of expression, that they do include in the rebuttal.

Response: We have explained the results of the control staining in the legend of the supplementary Figure 9b as suggested.